# Natural plant growth and development achieved in the IPK PhenoSphere by dynamic environment simulation

Marc C. Heuermann [1] ✉, Dominic Knoch [1], Astrid Junker[1,2] & Thomas Altmann [1]

In plant science, the suboptimal match of growing conditions hampers the transfer of knowledge from controlled environments in glasshouses or climate chambers to field environments. Here we present the PhenoSphere, a plant cultivation infrastructure designed to simulate field-like environments in a reproducible manner. To benchmark the PhenoSphere, the effects on plant growth of weather conditions of a single maize growing season and of an averaged season over three years are compared to those of a standard glasshouse and of four years of field trials. The single season simulation proves superior to the glasshouse and the averaged season in the PhenoSphere: The simulated weather regime of the single season triggers plant growth and development progression very similar to that observed in the field. Hence, the PhenoSphere enables detailed analyses of performance-related trait expression and causal biological mechanisms in plant populations exposed to weather conditions of current and anticipated future climate scenarios.

Plants express plastic phenotypes that are influenced by the constraints and dynamics of the environment with which they interact[1–4]. Thus, any phenotype of a plant is a function of its genotype (G) interacting with the environment (E), whether natural or artificial. The effects of the latter are often extended by the term management (M), and the result of all possible combinations can be condensed to the term phenome[5], which in turn means that each phenotype represents only one possible instance among the multitude of GxExM combinations.

A plant phenotype can be recorded or measured either outdoors in a native/target environment or indoors in a controlled environment. The former, field phenotyping, and the latter, controlled-environment phenotyping, are rapidly developing and improving disciplines driven by technical innovations, and both approaches have advantages and disadvantages, which have been reviewed extensively[6–10]. In the field, plants are exposed to natural environments varying by season and location and characterized by strong dynamics in air temperature and humidity, light quality and quantity, nutrient and water availability, soil composition and compaction and

weed and pest pressure[11]. Global climate change will further increase weather dynamics and will enhance the frequency and extent of weather extremes[12]. Indoors, e.g., in growth chambers and greenhouses, a climate for plant growth has to be created and maintained. This usually involves shifted mean levels and much lower amplitudes and dynamics of the environmental variables than in the field, such as light intensities and air and soil temperatures due to technical limitations or cost-efficiency[11]. Furthermore, in most indoor systems, plant performance is limited by pot size constraints[13]. Thus, natural outdoor environments are more relevant for assessing the expression of performance traits in crops since phenotypes measured on plants cultivated in greenhouses or climate chambers often deviate greatly from those observed for plants exposed to natural conditions in open fields, even considering common agricultural management practices. But the lack of control, i.e., the large and mostly unpredictable variation of natural conditions, can also be a strong disadvantage. The highly variable outdoor environment challenges the repeatability of experiments from season to season and often results in uncorrelated phenotypes due to random variation of weather

[1]Department of Molecular Genetics, Leibniz Institute of Plant Genetics and Crop Plant Research (IPK), Corrensstrasse 3, 06466 Seeland OT Gatersleben, Germany. [2]Present address: Syngenta Seeds GmbH, Zum Knipkenbach 20, 32107 Bad Salzuflen, Germany. ✉e-mail: heuermann@ipk-gatersleben.de

conditions[11]. For example, in a multi-environment study analyzing quantitative traits in maize, the vast majority of the forty-eight detected loci (QTL) showed contextual effects and were only found in specific environmental scenarios, except for only one common QTL that spans all environments[14]. For most indoor systems, the opposite is true. While the growth conditions can often be meticulously controlled and responses to treatments can be measured precisely, the phenotypes measured under those artificial interactions between genotypes and controlled environments can lead to low correlation and overestimations of trait effects in comparison to field experiments[15–17]. Despite the availability of facilities such as rain out shelters and free-air $CO_2$ enrichment (FACE) installations[18,19], simulating weather conditions expected to occur in future climate scenarios is nearly impossible in today's field experiments. Such simulations require that controlled environments be designed to account for dynamics of field conditions but also enable repeatable results[10].

In an effort to combine the relevance of natural dynamic environments and the repeatability of controlled-environment phenotyping, we designed and built the PhenoSphere (PS) at the Leibniz-Institute of Plant Genetics and Crop Plant Research (IPK) in Gatersleben, Germany, which we teased in a recent review[10]. In the PhenoSphere, plants can be grown under simulated field-like conditions in two large compartments, which can either be combined to merge the available space or operated independently. Inside, plants are cultivated in large-volume soil containers and a free air space of approx. six meters between the soil surface and the light sources so that all important crop species can develop fully. Weather variables like air and soil temperature, relative air humidity, vapor pressure deficit (VPD), light quality and quantity, $CO_2$ levels, and wind simulation can be automatically controlled and manipulated to mimic seasons, day length, day and night cycles, and field-like frequency and amplitude dynamics of conditions. In addition, it is planned to install a multisensory top-view imaging platform on an x-y-z positioning system for automated plant phenotyping.

In this study, we benchmarked the capabilities of the PhenoSphere to simulate field-like environments by continuously monitoring plant growth and developmental progression of eleven phenotypically diverse *Zea mays* inbred lines during their entire life cycles in the IPK field trial area (in four consecutive years) in a high standard climatized glasshouse (one season), and in the PhenoSphere. In the latter, two experiments were conducted, the first simulating average conditions of the years 2016–2018 and the second closely resembling the single season of the year 2016. By simulating the environment of one particular year, the maize plants in the PhenoSphere grew and developed at the same rate as in the field in terms of plant height, leaf stage and number, and flowering time and differed significantly from the phenotypes measured in the glasshouse or the PhenoSphere running an averaged climate regime. Thus, the PhenoSphere was indeed able to simulate a typical field-like growth environment and will allow users to analyze the growth performance of various plant species under diverse weather conditions corresponding to current and expected future climate scenarios.

## Results

The potential of simulating field-like environments, and thus of eliciting field-like expression of growth and developmental phenotypes, was evaluated inside the PhenoSphere and compared to that of field- and glasshouse-grown maize plants (Fig. 1). Phenotypes were recorded continuously throughout the plants' life cycles of a population of 11 phenotypically highly diverse *Zea mays* inbred lines with origins in North America, Europe, and Asia (Supplementary Table 1), which were selected to cover a wide range of sizes and developmental characteristics including flowering. All code developed for the subsequent analyses is provided in Supplementary Code 1.

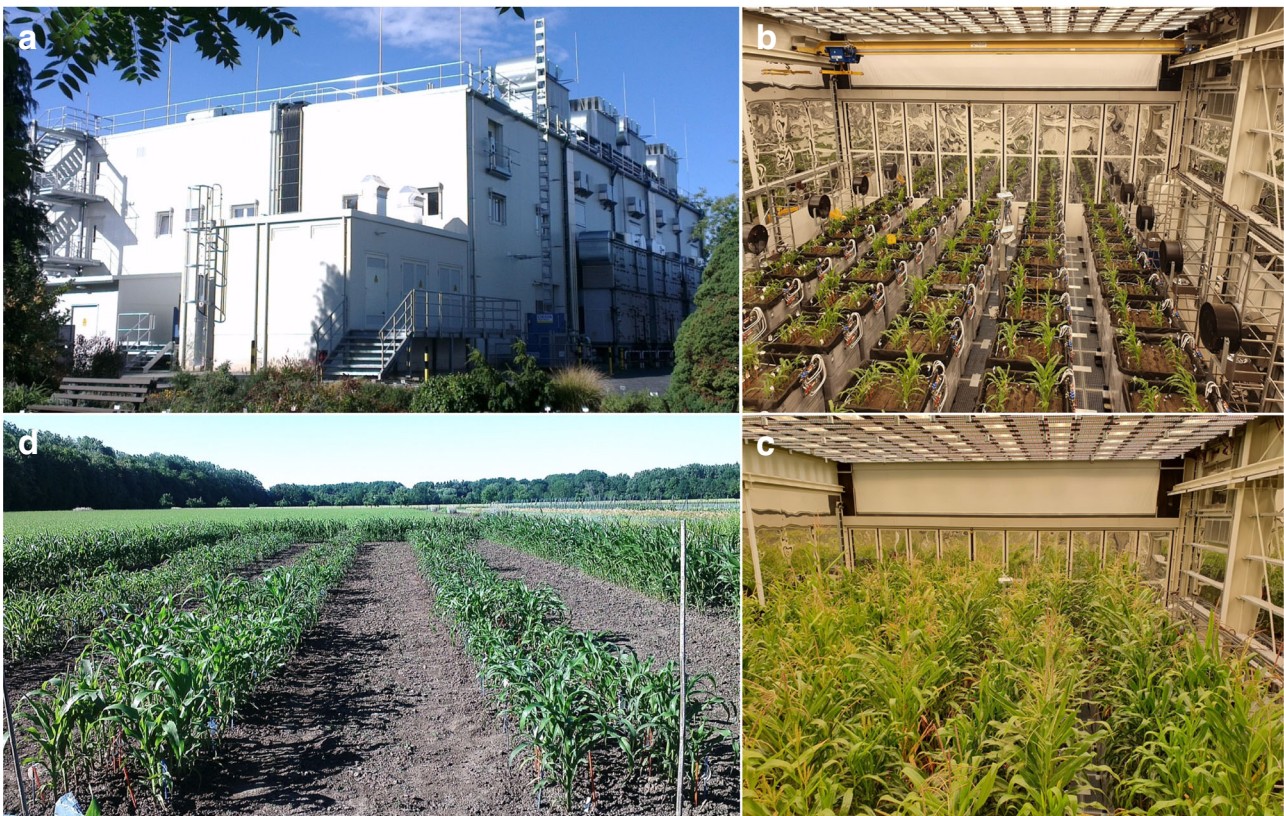

**Fig. 1 | The experimental sites at the IPK Gatersleben. a** The PhenoSphere as seen from the outside. View into compartment 1 of the PhenoSphere early (**b**) and late (**c**) in the growth period. **d** The field 2016 at the IPK field site.

## Weather simulation

To collect benchmark data for the phenotypes and the environments, maize plants were phenotyped at the IPK field sites in the four cultivation seasons from 2016 to 2019. The climatization regimes run in the PhenoSphere were programmed from real-world measurements of weather variables, including air temperature, relative air humidity, and global radiation, which were recorded at the IPK weather station and aggregated hourly during the four growing seasons (Supplementary Data 1). For the averaged climate experiment in the PhenoSphere ('PhenoSphere avg' experiment), the weather regime was reproduced from the seasons 2016, 2017, and 2018.

For each calendar week to be modeled, the 21 days from the three seasons were used to determine the seven daily median days. The median day with the lowest day temperature profile of a calendar week over all three seasons was deemed to be the cloudy day, and likewise, the day with the highest temperature profile of that calendar week was regarded the sunny day. The 24-h profile of hourly averaged temperatures over all 7 days of the median week of each calendar week across the 3 years was selected to represent the normal day. Each week of the experiment was simulated independently and consisted of three normal days, two cloudy days, and two sunny days. Their consecutive occurrence was set once, and the pattern was kept for successive weeks.

In a subsequent experiment in the PhenoSphere ('PhenoSphere 2016 sim' experiment), we aimed to simulate the characteristic weather regime of a single growing season using 2016 as template. Normal days were still averaged over one week, as described above, but only from the 2016 growing season. The sunny and cloudy days were modeled, recreating the temperature profiles of uniquely measured days following the same criteria with the highest and lowest day temperatures. In contrast to the 'PhenoSphere avg' experiment, the order of the days and their number was adapted weekly to closely follow the real-world template. An experiment in a standard climatized glasshouse with pot cultivation was used as a reference for a controlled-environment. In all environments, the temperature was logged in hourly intervals over the whole growing season (Fig. 2). The light conditions (see Supplementary Tables 4–6 for the program and Supplementary Fig. 19 for spectra) and relative humidity values were not reproduced daily but depended on the three model days. Three light regimes were adjusted weekly according to the time of sunrise and sunset. Beginning at sunrise, relative humidity decreases linearly from 90% at night to 40% (normal/sunny days) and 50% (cloudy days), with humidity levels remaining stable for the central 5 h during the light period. After the end of the central light period, the relative humidity increased linearly to 90% by sunset.

To assess the degrees of match between the real world and the programmed controlled weather regimes, pairwise correlations of the measured temperature profiles of all seven scenarios were calculated. The highest correlation was observed between the temperature profiles of the 'PhenoSphere 2016 sim' experiment and the 2016 cultivation season itself ($r = 0.88$, $R^2 = 0.77$, $p = < 0.001$; Fig. 3). Lower correlations were detected between 'PhenoSphere avg' with either the growing season 2016 or its simulation (Fig. 3) but the correlation between 'PhenoSphere avg' and its template years was slightly higher than among the template years themselves (Supplementary Fig. 1). The correlations between the field environments 2017, 2018, and 2019 with both 2016 and 'PhenoSphere 2016 sim' followed a comparable trend with lowest measured against 2017 and highest against 2019 (Supplementary Fig. 1). The temperature profile of the glasshouse appeared to be most divergent and correlated lowest with any dynamic environment (Supplementary Fig. 1). The root mean squared errors (RMSE) of the hourly temperature profiles (Fig. 2) relative to the field 2016 was lowest against the 'PhenoSphere 2016 sim' (Supplementary Table 2).

The profiles of the hourly calculated vapor pressure deficit (VPD) values derived from the sensor data (Supplementary Data 1) of the seven environments (Supplementary Fig. 2), which were also most strongly correlated between the field 2016 and the 'PhenoSphere 2016 sim' ($r = 0.87$, Supplementary Fig. 3), had lowest RMSE between field 2016 and the 'PhenoSphere 2016 sim' (Supplementary Table 2).

The comparisons of the environments were complemented using the concept of thermal time, which determines the species-specific temperature response to growth as a ratio relative to developmental rates under a hypothetically constant 20 °C environment[20]. The thermal time day profiles (Supplementary Fig. 4) were again most similar in the 'PhenoSphere 2016 sim' and the field 2016 and were most strongly correlated ($r = 0.85$, Supplementary Fig. 5), while only low correlations were found among any of the other environment comparison (Supplementary Fig. 5). The RMSE between the thermal time day profiles relative to the field 2016 was lowest for the 'PhenoSphere 2016 sim' (Supplementary Table 2).

## Plant growth performance and developmental progression

Plant height and developmental stage were used as traits to assess the effects of the different environments on plant growth and developmental trajectories (Supplementary Data 2). Phenotyping was done manually/visually twice a week throughout the entire cultivation period, and trait expression over time was found to be most field-like in the 'PhenoSphere 2016 sim' experiment. Lower match was found in the 'PhenoSphere avg' experiment, and the largest deviation from the growth and development progression in the field cultivations occurred in the glasshouse experiment. The appearances of the plants are exemplarily shown for the lines ZEA 851, ZEA 332, and B73 at around 48 days after sowing (DAS) in Fig. 4. At 48 DAS in 'PhenoSphere 2016 sim' the plants were just slightly taller than in the field 2016 and V-stages were comparable (Fig. 4). In the 'PhenoSphere avg' experiment, plants of the same age were already twice as tall but were still surpassed in size by plants in the glasshouse. Furthermore, in both environments, V-stages were further advanced than in any field trial or in the 'PhenoSphere 2016 sim' experiment (Fig. 4).

The goal of the experiments was to assess the degree of matching of the phenotypes expressed in each environment based on plant growth and development progression during all phases of the growth period of the population of highly diverse maize lines. Thus, the measured traits were analyzed using nonlinear logistic growth models by fitting sigmoidal curves to the time-dependent progression of plant height and developmental stages for each environment. The models estimated the maximal endpoint values, the inflection points, which represent the point of maximal growth speeds, and the steepness of the curves or the growth rates.

A difference in final population plant height between the field cultivations was significant only between 2016 and 2019 (Fig. 5). Similarly, the plant height reached in the 'PhenoSphere 2016 sim' experiment was also significantly larger than in the field-grown plants in 2019 (Fig. 5). The 'PhenoSphere avg' and the glasshouse experiments produced the tallest plants with heights significantly larger than those of the plants in the 2017, 2018, and 2019 field cultivations. With the exception of the field experiment 2016, the plants in the 'PhenoSphere avg' were significantly taller than the plants in the other template environments 2017 and 2018 and their heights were not significantly different from that of the plants in the glasshouse (Fig. 5).

The strongest discriminatory parameter of the plants in the various environments was the inflection point (xmid) of the sigmoid curves, the time point at which the maize population as a whole grew fastest in the respective environment (Fig. 5, Supplementary Fig. 6). The field environments and the 'PhenoSphere 2016 sim' evoked the maximum growth rate at about 68 DAS and the differences among these five environments were not significant (Fig. 5). Plants in the glasshouse environment grew significantly faster and reached the inflection point about 30 days earlier than the plants in the fields and in the 'PhenoSphere 2016 sim' (Fig. 5). The averaged environment in 'PhenoSphere avg' also led to significantly faster growth than in the

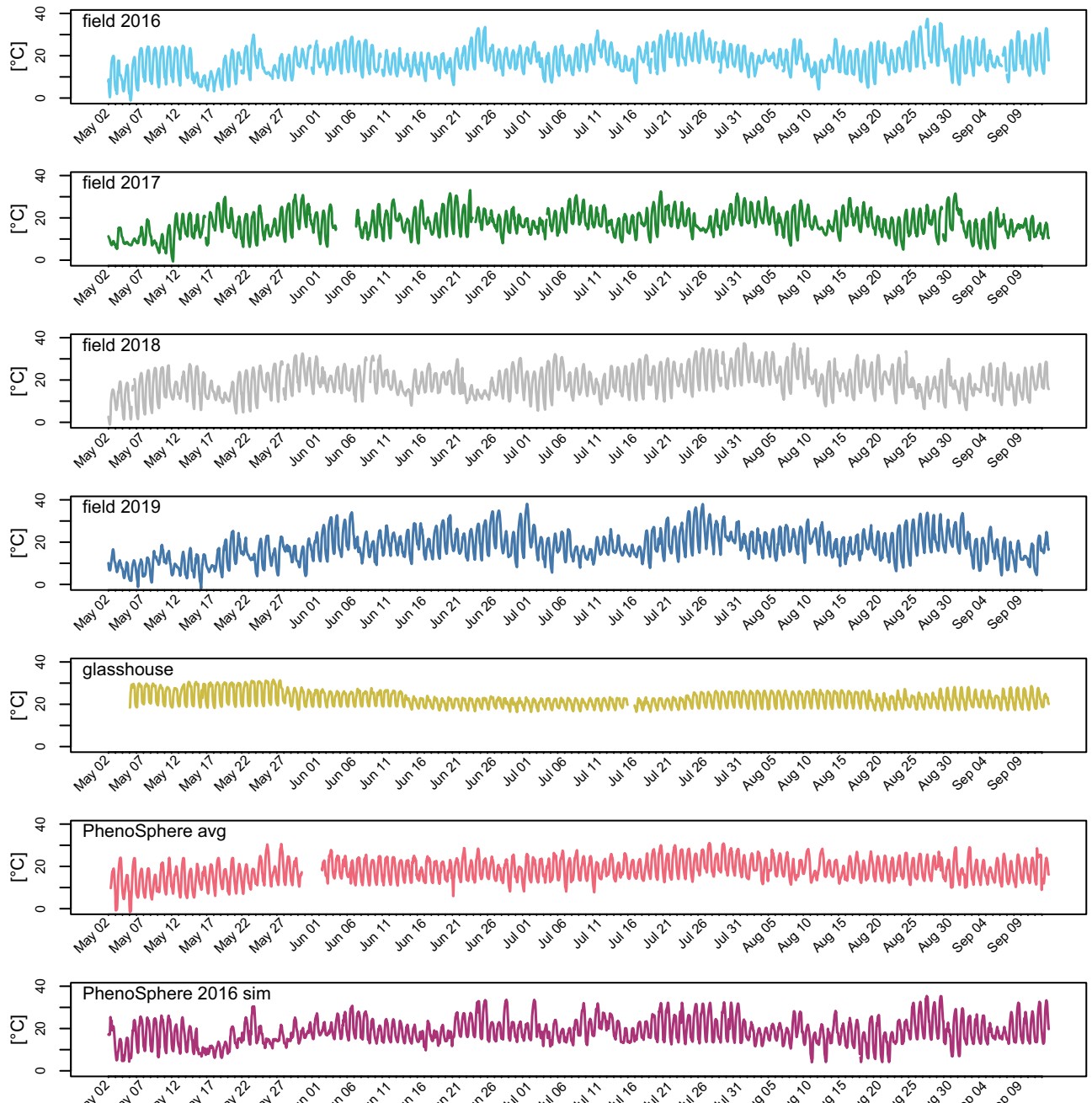

**Fig. 2 | The temperature profiles of the seven conducted field and indoor experiments.** Temperature profiles of the field 2016–2019 (2nd May–13th September), of the glasshouse, of the 'PhenoSphere avg', and the 'PhenoSphere 2016 sim' experiments. Temperature values were plotted in an hourly resolution except for minor gaps due to sensor failures. Dates for the field experiments are true calendar dates, and the dates in the controlled environment have been relatively mapped to simulate their templates. Colors used: field 2016 as Sky Blue (#66CCEE), field 2017 as Forest Green (#228833), field 2018 as Silver (#BBBBBB), field 2019 as San Marino (#4477AA), glasshouse as Turmeric (#CCBB44), PhenoSphere avg as Froly (#EE6677), and PhenoSphere 2016 sim as Royal Heath (#AA3377).

fields or in the single year simulation ('PhenoSphere 2016 sim') and the inflection point was reached about 20 days earlier than in any field season or in the 'PhenoSphere 2016 sim' (Fig. 5). The difference between the inflection points of the 'PhenoSphere avg' and the glasshouse environment was not significant (Fig. 5).

At the average inflection point for the field and 'PhenoSphere 2016 sim' experiments, xmid at 68 DAS, plant height values were calculated from the modeled curves for each individual genotype in each environment (Supplementary Fig. 9). Plant height at an xmid of 68 DAS was clustered and a closest relationship was found between the genotypic performance between the 2016 field and the 'PhenoSphere 2016

sim'. They clustered together with the other field years in an own clade and were distant to the 'PhenoSphere avg' and the glasshouse environment, which formed another clade (Supplementary Fig. 7).

Regardless of the environment, the plants reached similar final developmental stages; the final V-stage did not differ (Fig. 6). However, temporal developmental progression was different in the 'PhenoSphere avg' and the glasshouse, where leaves matured significantly earlier than in the field or in the 'PhenoSphere 2016 sim' experiments (Fig. 6, Supplementary Fig. 8). No significant differences could be detected between leaf maturation speed in the 'PhenoSphere 2016 sim' experiment or in any of the field environments (Fig. 6). The trait

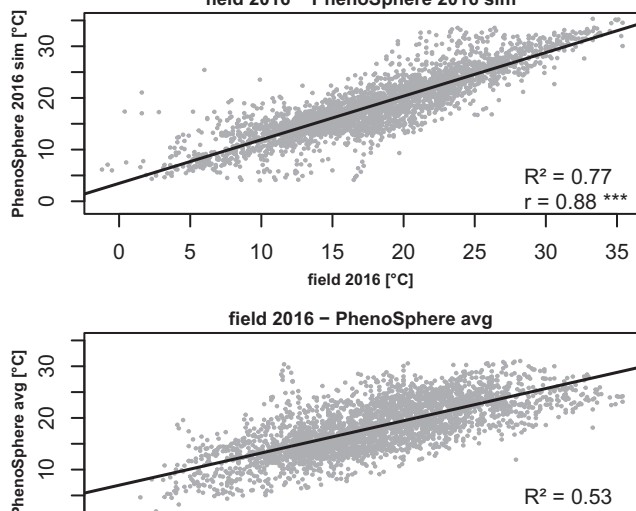

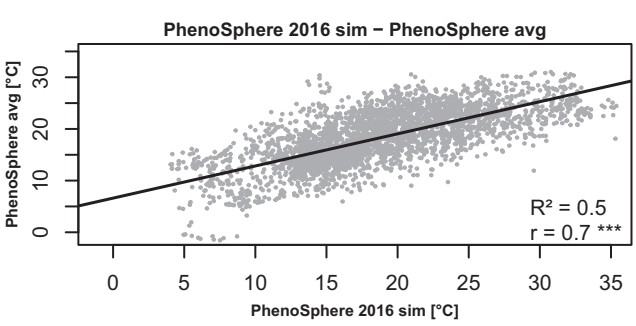

**Fig. 3 | Correlations of temperature values in the field 2016 experiment and the two PhenoSphere experiments.** Pairwise correlation plots of the temperature profiles over the whole duration of the growing season of the field 2016, the 'PhenoSphere avg', and the 'PhenoSphere 2016 sim' experiments. Pearson correlation coefficients are noted as 'r', their significance threshold (two-sided) of $p < 2.2\text{e}{-}16$ is indicated by three asterisks, and the coefficient of determination 'R²' values are derived from squared 'r' values.

growth stage, in which the appearance of all leaves was tracked, followed the same pattern (Supplementary Figs. 9 and 10).

Modeling the plant height versus thermal time days showed the same pattern, with the inflection point of the 'PhenoSphere 2016 sim' not differing from the field environments (Supplementary Fig. 11). Plants in the glasshouse and the 'PhenoSphere avg' developed significantly faster than any field environment or the 'PhenoSphere 2016 sim' (Supplementary Fig. 11). Interestingly, the positions of the glasshouse and the 'PhenoSphere avg' switched and the plants in the 'PhenoSphere avg' grew fastest when scaled to thermal time days. For the trait plant height, the repeatability of the genotypes was 0.55 and was approximated from a linear mixed-effect model over all time points and all environments with a conditional $R^2$ of 0.96 and a marginal $R^2$ of 0.87.

Considerable genotype x environment interaction was detectable in the different environments, to some extent even in the direct comparison between the field 2016 and the 'PhenoSphere 2016 sim', but the individual genotypes followed the overall trend of the population mean (Supplementary Figs. 12–15).

## Modeling of flowering time
The occurrence of tassels was tracked for each phenotyped plant as a discrete value of either presence or absence (Supplementary Data 2).

The tassel occurrence was converted into a population-wide percentage of plants having a tassel at each time point. Thus, the data to be modeled had no variation at each DAS, and a fixed-effect nonlinear regression model was fit to the data to distinguish between the environments on a whole population level (Fig. 7). The cultivation scenarios and time were treated as fixed factors and found to have significant effects. In the field environments and the 'PhenoSphere 2016 sim' tassels occurred about 20 and 30 days later than in either the 'PhenoSphere avg' or the glasshouse, respectively (Fig. 7, Supplementary Fig. 16).

### Yield components
For a more comprehensive documentation of the experiments, yield components like number of ears, grain number, total grain weight and average individual grain weight were recorded (Supplementary Data 2). Individual grain weight was consistent in all environments and total grain weight per plant was similar for plants grown in the field experiments in 2016/17/18, in the glasshouse, and in the 'PhenoSphere avg' experiment, but reduced grain numbers and thus reduced total grain weights per plant were observed in the 'PhenoSphere 2016 sim' experiment (Supplementary Fig. 21). The number of ears was elevated in the 'PhenoSphere 2016 sim' in comparison to the field environments but was unchanged for the other pairwise environment combinations (Supplementary Fig. 21).

### Homogeneity test experiment
The PhenoSphere was designed to homogeneously control the environmental conditions within the PhenoSphere cultivation area, which was checked with a separate experiment. The experiment aimed to decompose the variance of the experimental design components such as the effect of the soil treatment, of the container position (row and column), of the replicates and of the potential effect of the pre-cultivated genotypes in the reused soil. Hierarchical variance decomposition with *rptR*[21] showed that throughout the experiment, from 21 to 70 DAS, between 0.6 and 0.8 of the total variance was attributed to the genotype effect and between 0.15 and 0.3 to the residual error (Supplementary Fig. 17, Supplementary Table 3). After the plants matured and desiccated between 77 and 90 DAS, the variance contribution of genotype and residuals reversed. The effects of column, replicate, and soil treatment were never contributing significantly. The effect of the pre-cultivated genotype contributed with 0.02 at two time points (27, 49 DAS) and the effect of the row of containers between 0.03 and 0.06 at three time points (21, 27, 49 DAS) toward the total measured variance (Supplementary Fig. 17, Supplementary Table 3). Thus, we concluded that the controlled environmental conditions are indeed spatially highly homogeneous across the cultivation areas of the PhenoSphere.

## Discussion
The weather simulation, derived from three representative model days, proved sufficient to match the natural environment and resulted in field-like plant growth and developmental progression in the simulation of a single field season. The high spatial homogeneity of the PhenoSphere exposed all plants in the containers equally to simulated environmental regimes. The correlation between the weather simulation and the outdoor environment concerning temperature, thermal time, and VPD profiles over the cultivation periods was highest when using real days as templates in the single season simulation, which was supported by the lowest RMSE, and also considering their order and frequency in a given week. The daily cumulative thermal time contribution of individual calendar days would only correlate between environments if the order and frequency were respected, like between the single season simulation and the field 2016 (Supplementary Fig. 5). By choosing the median days of three vegetation periods in the 'PhenoSphere avg' experiment, always the most moderate day was chosen,

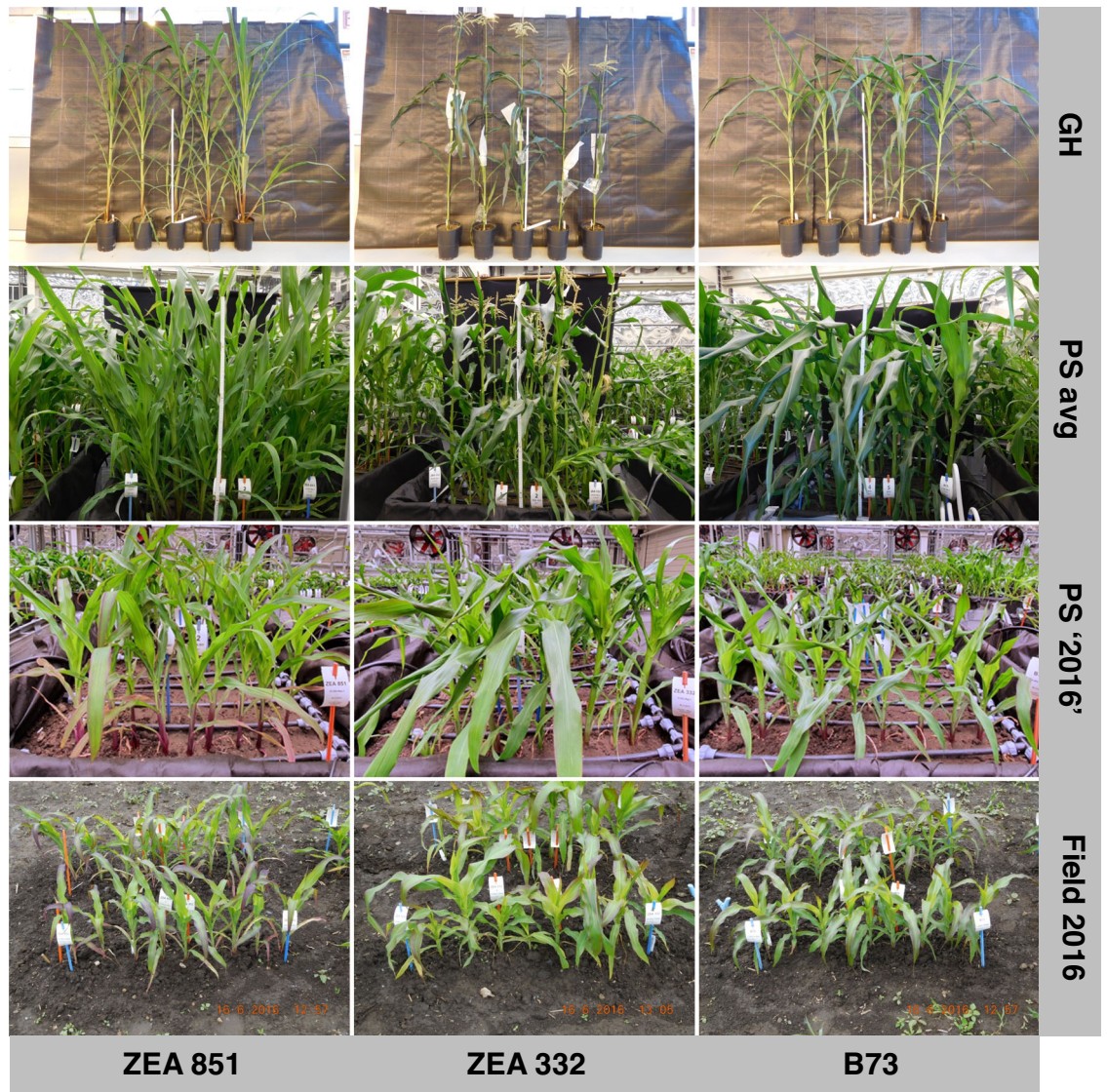

**Fig. 4 | Habitus of plants of the same age grown in different indoor and field environments.** Example images of the three genotypes ZEA 851, ZEA 332, and B73 taken in the glasshouse (GH) [47 DAS, 2018-11-28; 169 cm (V9), 172 cm (V10), and 142 cm (V10) respectively], in the 'PhenoSphere avg' (PS avg) [47 DAS, 2018-11-28; 123 cm (V8), 123 cm (V11), and 111 cm (V9) respectively], in the 'PhenoSphere 2016 sim' (PS '2016') [48 DAS, 2020-01-23; 53 cm (V6), 55 cm (V6), and 42 cm (V6) respectively], and in the field 2016 [50 DAS, 2016-06-16; 39 cm (V7), 36 cm (V6), and 36 cm (V7) respectively]. The provided values are the BLUEs of plant height and V-stage.

which together with a fixed frequency led to a lower amplitude of condition variation, fewer extreme values and thus to a lower correlation with the values measured in the field. Future simulations of target weather regimes could be further improved by determining the most representative daily weather profiles from several years of historic weather data, e.g., with clustering algorithms, and derive their frequencies of occurrences from the cluster sizes. The temperature ranges currently achieved in the PhenoSphere will, however, constrain locations and seasons to be simulated to instances with temperatures above 0 °C and therefore limit the simulation of late autumn, winter, or early spring when outside temperatures can drop below 0 °C.

Simulating a single maize growing season within the PhenoSphere and using large-volume soil containers resulted in plant growth and development progression that did not differ significantly from the rates that the same population exhibited upon cultivation in the corresponding field season. Field-grown plants and plants inside the 'PhenoSphere 2016 sim' experiment required the same amount of time to reach peak growth speed, maturity of leaves, and tasseling. This made the effects of the field and field-like environment of the

'PhenoSphere 2016 sim' very different from the 'PhenoSphere avg' experiment, where, despite its dynamic environment, the plants developed much faster and even faster under thermal time.

Averaging weather variables resulted in phenotypes more intermediate between field-grown plants and plants grown in the climatized greenhouse, in which plants expressed the fastest growth and development rate, thus proving to be the most artificial environment. The difference between the 'PhenoSphere 2016 sim' experiment and the 'PhenoSphere avg' and glasshouse experiments were as evident after the conversion of the temperature profiles into thermal time days, which transforms growth progression into a new temporal dimension. This is basically a scaling according to temperature and emphasizes the remaining differences in feature expression due to other environmental factors such as light and/or relative air humidity (VPD) regimes.

The tight clustering of the 'PhenoSphere 2016 sim' with the field environments for the plant heights of the individual genotypes, especially with the year 2016, at the average inflection point for the fields and field-like environments supported the observed field-like

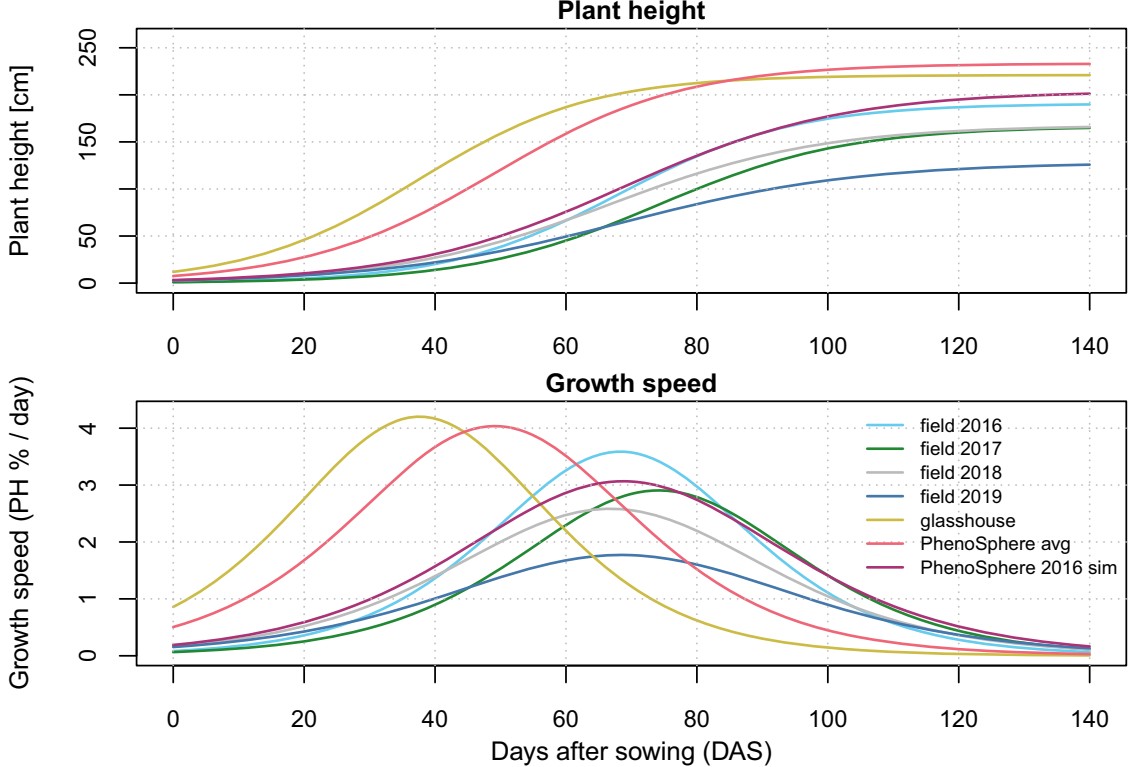

**Fig. 5 | Plant heights and growth speeds over time observed in seven field and indoor environments.** Plant height plotted as a best fit logistic growth curve (for fit to BLUEs, see Supplementary Fig. 6) representing the growth trend of the maize population in each environment versus days after sowing (DAS). The first derivative of the fitted curve represents change in plant height in percent per DAS for each environment. Colors used: field 2016 as Sky Blue (#66CCEE), field 2017 as Forest Green (#228833), field 2018 as Silver (#BBBBBB), field 2019 as San Marino (#4477AA), glasshouse as Turmeric (#CCBB44), PhenoSphere avg as Froly (#EE6677), and PhenoSphere 2016 sim as Royal Heath (#AA3377).

development within the single season simulation also at the individual genotype level (Supplementary Fig. 7, Supplementary Figs. 12–15).

Although the focus of this study was on vegetative growth traits up to flowering, yield parameters such as number of ears and total number and weight of grains per plant were recorded for each genotype in each environment to document the experiments. In the benchmark environment in the field 2016, the 11 genotypes were part of a larger cultivation set with 30 genotypes, while in the 'Pheno-Sphere 2016 sim', only the 11 selected genotypes were cultivated. The longer vegetation period in the 'PhenoSphere 2016 sim' and the corresponding wider spread of flowering times meant that compared to the greenhouse and to the 'PhenoSphere avg', fewer genotypes released pollen at the same time. Because the genotypes were grown in double-row plots with random genotypes as neighbors, the next container of the same genotype ($n = 5$) was usually too far apart to serve as pollen donor. Thus, most probably too low pollen availability in the in 'PhenoSphere 2016 sim' experiment resulted in lower grain number and reduced total grain weight per plant. The increased number of ears in 'PhenoSphere 2016 sim' hints at potential compensatory responses of the plants to the low seed set (Supplementary Fig. 21). Besides the consequences of the wide range of flowering times in the investigated maize population, environmental parameters such as the total intercepted light, or the night temperatures or soil water potential during flowering have been shown to affect grain number in a genotype-dependent manner[22] and may have contributed to the differences between the 'PhenoSphere avg' and 'PhenoSphere 2016 sim' experiments. Interestingly, the observed effects on grain yield and yield components were genotype-dependent with different genotype x environment interactions (Supplementary Fig. 22).

As shown in the 'PhenoSphere avg' experiment, maize plants can fully mature and produce grains in the PhenoSphere in comparable quantities as in field trials. For a proper evaluation of grain yield formation, however, a very different experimental setup will be required, either by using genotypes with a very narrow range of flowering times and/or by creating larger plots composed of multiple adjacent containers of the same genotype and by assessing the yield in the central container(s). Furthermore, the aforementioned environmental parameters need to be considered and carefully set in experiments designed to evaluate grain yield in the PhenoSphere.

The presented results show that in the PhenoSphere a field-like environment can be simulated in terms of the evoked plant growth rates and the progression of developmental stages, resulting in field-typical plant growth performance. The PhenoSphere thus fills the gap between hitherto established controlled-environment phenotyping systems and field phenotyping trials[10]. Compared to other phenotyping platforms, the PhenoSphere occupies a unique position by enabling the exposure of crop plants to repeatable and specifically designable dynamic environments that mimic relevant natural conditions, ranging from benign to detrimental[6,9–11]. The ability to elicit field-like growth and development in the dynamic but controlled environment of the PhenoSphere is a very substantial and important advance and goes far beyond previous improvements in standard climatized glasshouse cultivation procedures with which correlation with the field was increased by imposing adjusted temperature regimes[11,23] The technical capabilities of the PhenoSphere overcome several limitations of typical growth chambers and glasshouses[11,17] such as light intensity level and variation, light spectrum (including the visible range and UV-A but no UV-B), air temperature, humidity, and VPD ranges and variation enabling extremely high levels (albeit no frost), $CO_2$

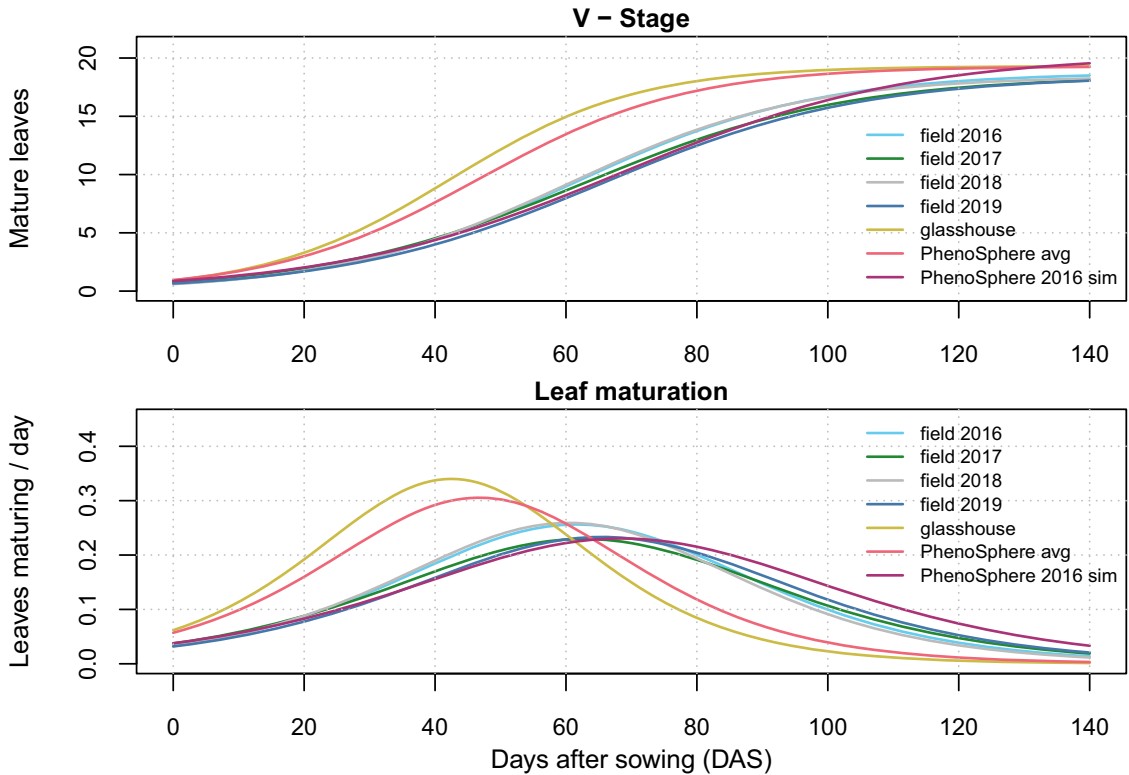

**Fig. 6 | Developmental stage progression and leaf maturation rates over time observed in seven field and indoor environments.** Vegetative stage (V-stage) plotted as a best fit logistic growth curve (for fit to BLUEs, see Supplementary Fig. 8) representing the maturation of leaves of the maize population in each environment versus days after sowing (DAS). The first derivative of the fitted curve represents change in leaves maturing per DAS for each environment. Colors used: field 2016 as Sky Blue (#66CCEE), field 2017 as Forest Green (#228833), field 2018 as Silver (#BBBBBB), field 2019 as San Marino (#4477AA), glasshouse as Turmeric (#CCBB44), PhenoSphere avg as Froly (#EE6677), and PhenoSphere 2016 sim as Royal Heath (#AA3377).

concentration (regulated elevation), and wind simulation. The container system even overcomes limitations of soil volumes in pots and enables the use of field-like soil structure and composition, controlled (drip) irrigation, and soil temperature control independent from air temperatures. The optimized and validated field-like environment simulation programs can now be used to perform also (grain) yield trials, which require a specific experimental setup and designs fundamentally different from the benchmarking experiments of this study.

Clear limitations of the PhenoSphere are biotic interactions. Topsoil taken directly from the field can aid in microbial interaction, while interaction with insects or animals is limited, which can be a strength but is also artificial.

We present the PhenoSphere to the plant science community as a novel tool to study the plant's response to variation in weather variables and environmental conditions specifically tailored to the addressed research topic: Deliberate temperature profiles can be realized with an hourly resolution, light quality and quantity can be manipulated on a minute resolution scale, wind speed and direction can be changed on the sub-hour scale, atmospheric $CO_2$ levels can be increased, water and fertilization can be automatically applied daily, and the large-volume containers allow the use of different soil types and compositions and the modulation of the soil temperature.

For fundamental and application-oriented research, there is a need to expose plants, especially crop plants, to relevant field-like conditions in a reproducible manner. This will support systems biology analyses carried out to elucidate molecular mechanisms underlying the expression of agronomically relevant traits. It furthermore enables testing hypotheses derived from approaches such as network analysis and modeling on the consequences of genetic variation and on the propagation of the elicited effects through biological pathways/networks that result in variation of performance-related trait expression under a range of different environmental conditions. Finally, the PhenoSphere allows stakeholders to study the performance of specific genotypes under beneficial and detrimental weather conditions expected to occur in future climate scenarios (compared to contemporary situations).

## Methods

### PhenoSphere technical specifications
The PhenoSphere has a ground layout of $40 \times 25$ m with an eave height of 10.5 m and harbors four compartments of equal size (each 189 m² ground area) and a work and pre-/post-processing area of 276 m² (Supplementary Fig. 18). A large rhizotron system for root phenotyping is integrated into compartments 3 and 4, which will be presented in detail elsewhere. A container-based cultivation system is installed in compartments 1 and 2, which are fully climatized and allow field-like environment simulation and were used in this study. Up to 55 containers per compartment are positioned onto a steel grating floor 1 m above the ground floor. From the assimilation light sources (8.09 m above ground floor) to the edge of the container (2.3 m above ground floor), 5.77 m of air space (c. 5.90 m from soil surface to lamps) is available for the plant to develop fully and still have sufficient distance to the lamps. One air ventilation system dispenses conditioned fresh air by air hoses below the steel grating floor and takes out air through exhaust vents 6 m above ground level. Five large fans on either side of the cultivation area in each compartment enable wind simulation with up to 6 m/s (4 Bft). A second air circulation and cooling system is deployed to cool the light source. Exhaust vents directly below the lights prevent heated air from the lamps from mixing with the conditioned air from the main cooling system. The upper temperature limit

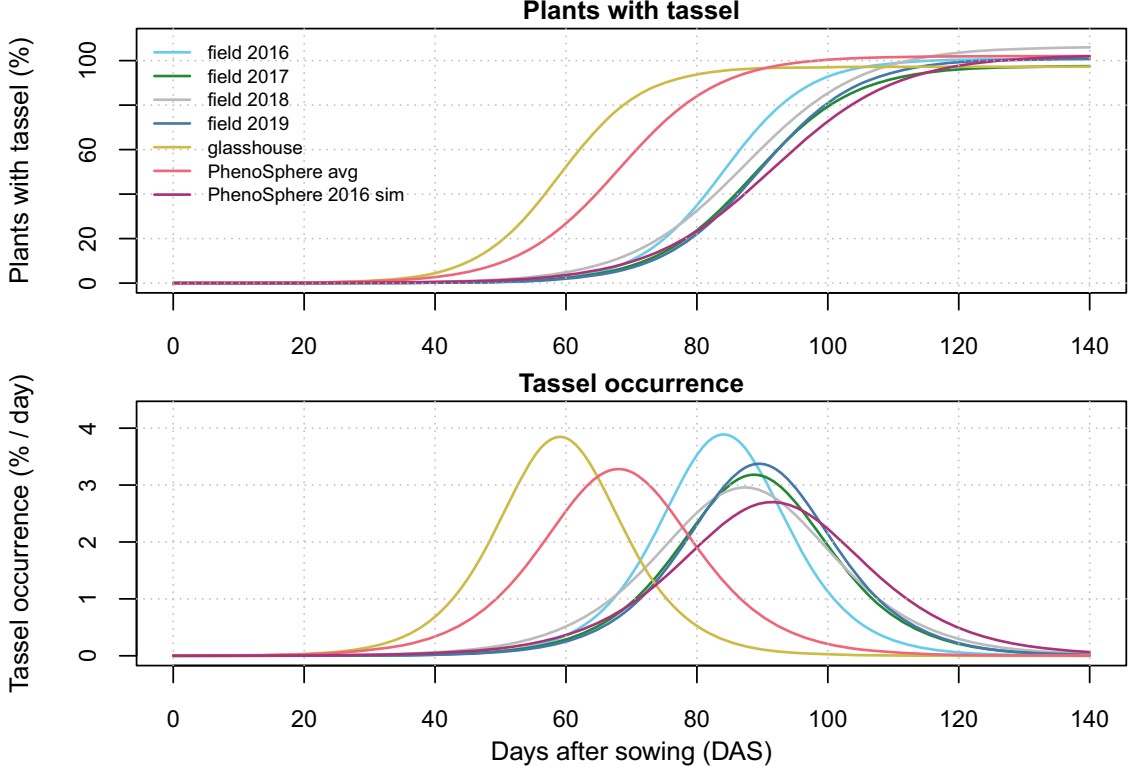

**Fig. 7 | The pace of tassel formation in seven field and indoor environments.** Percentage of plants of the maize population with tassel present plotted as a best fit logistic growth curve (for fit to data, see Supplementary Fig. 16) of the maize population in each environment versus days after sowing (DAS). The first derivative of the fitted curve represents change in the percent of the plants with tassel occurrence per DAS for each environment. Colors used: field 2016 as Sky Blue (#66CCEE), field 2017 as Forest Green (#228833), field 2018 as Silver (#BBBBBB), field 2019 as San Marino (#4477AA), glasshouse as Turmeric (#CCBB44), PhenoSphere avg as Froly (#EE6677), and PhenoSphere 2016 sim as Royal Heath (#AA3377).

in compartments 1 and 2 is 47.5 °C at 35% relative humidity (VPD 6.94 kPa) and 37.5 °C at 7.5% relative humidity (VPD 5.99 kPa). The lower temperature limits depend on light intensities and the freezing point (>0 °C).

Assimilation light sources are composed by four independently controlled groups of General Electric Lighting CMH400 light bulbs (General Electric Deutschland Holding GmbH, D-60313 Frankfurt am Main) complemented by six different LEDs in intervening light bars (cool white 5700 K L1C1-5770, deep red L1C1-DRD1, far red L1C1-FRD1, blue L1C1-BLU1, royal blue L1C1-RYL1, and cyan L1C1-CYN1; LUMILEDS, D-52068 Aachen). Ultraviolet radiation (UVA 315–400 nm) is provided by NARVA LT 36 W T8/ 010 UV (NARVA, D-09618 Brand-Erbisdorf). All spectra of all light sources were individually recorded at 100% intensity and in various combinations (all 100%, sunny, normal, and cloudy day settings) using the same settings with an Ocean Optics USB2000+XR1-ES (Ocean Insight, D-73760) (Supplementary Fig. 19).

Total light intensity at 100% of all light sources was measured with a Licor LI-250A light meter coupled with a LI-190R-BNC-2 quantum sensor (LI-COR Bioscience GmbH, D-61352) at 20 cm, 120 cm, and 220 cm above soil level with averages over the cultivation area of 1277, 1348, and 1436 µmol m$^{-2}$ s$^{-1}$ PAR, respectively (Supplementary Fig. 20).

**Field cultivation**

Five *Zea mays* inbred lines from the yellow dent, stiff stalk, and non-stiff stalk breeding pools (B73, N22, P148, PHT77, and S052) and six accessions (amplified after single seed descent, SSD, passage) from the Genebank of the IPK Gatersleben (ZEA 132, ZEA 324, ZEA 332, ZEA 3660, ZEA 399, and ZEA 851) were grown in double-row plots in two replicates (three replicates in 2019) in a randomized block design, in total 22 double-row plots, in the field site at the IPK Gatersleben from April to September 2016/17/18/19 (for more information about the *Zea*

*mays* lines see Supplementary Table 1). In 2016/17/18, the fields were part of a larger cultivation with 30 genotypes in total. The stands in all years were protected by border planting of commercially available maize hybrids to mimic field conditions. Each double-row plot consisted of 18 plants, nine in each row, with dimensions of 80 cm length and 60 cm width. The plant-to-plant distance in one row was 10 cm, and the distance between rows was 60 cm. From the inner six individuals of each double-row plot, five representative individuals were selected and marked with an identifier, a label on the ground. These 110 plants were visually rated twice a week (Mondays and Thursdays) for plant height, growth stage and vegetative stage[24], time point of tassel emergence. The first batch of seed stocks for the experiment in 2016 was kindly provided by the German plant phenotyping network (DPPN) project as part of a nested reference collection. The seed stocks of the later experiments were self-propagated by line maintenance from the previous cultivations.

Seeds were sown out in Jiffy strips (5 × 5 × 5 cm bio-degradable pots, Hermann Meyer KG, D-01683 Nossen) filled with substrate 2 (Easy Growing, Klasmann-Deilmann GmbH, D-49744 Geeste) and cultivated under 16-h days at 24 °C/18 °C day/night in a greenhouse with auxiliary illumination for 7 days. Young seedlings were then transferred to a protected outside soil bed, which was protected from animal pests, to acclimate to outdoor climate conditions. After 4 days, plants were brought to the field site in the early morning hours (7 am) and transplanted together with the well-watered Jiffy into the field at a depth of 5–7 cm to fully cover the Jiffy. Initially, the field was sufficiently watered to ensure that plants are able to penetrate their roots through the bio-degradable Jiffy pots. Supplementary irrigation was applied in the following two weeks if necessary. Fertilizer was applied about one month after transplanting each year. In 2016, an ammonium sulfate fertilizer with 21% nitrogen and

24% water soluble sulfur was given and in 2017, 2018, and 2019 the fertilizer Nitrophoska 15+15+15 (+2S) 15% Nitrogen, 15% $P_2O_5$, 15% $K_2O$, 2% S (EuroChem Agro GmbH, D-68165 Mannheim) was applied to reach 75 kg $P_2O_5$ ha$^{-1}$.

## Glasshouse cultivation, PhenoSphere cultivations

In a parallel experiment (2018/19) in the PhenoSphere and in a large climatized glasshouse, the aforementioned lines were grown in double-row plots for 115 days in a container-based system and in single pots, respectively. Plants for both experiments were sown out and pre-cultured together in the same greenhouse used for the pre-treatments of field experiments and under the same conditions for 7 days. Subsequently, plants were transferred into the PhenoSphere ('PhenoSphere avg' experiment) and the glasshouse for a 4-day acclimatization period, after which they were transplanted into large-volume soil containers and single pots, respectively. In the 'Pheno-Sphere 2016 sim' experiment that only simulated one vegetation period (2016) and which was carried out in 2019/20, the plants were pre-treated in the same way as in the aforementioned 'PhenoSphere avg' experiment.

In the 'PhenoSphere avg' and 'PhenoSphere 2016 sim' experiments, the double-row plot layout per genotype of the field was applied to the containers, which led to five containers per genotype (each with a double row of 2 × 9 plants) and to a total of 55 containers for the 11 genotypes. Each container had a surface area of one square meter and a soil depth of 100 cm, which equals a volume of one cubic meter or 1000 L of soil volume. Containers were filled with layers of different materials to mimic a field-like soil-layer structure. The lowest layer (2.5 cm) was made up of floor elements with honeycombs filled with expanded polystyrene granules covered by a mat of fibers to enable drainage and prevent waterlogging and at the same time to serve as thermal insulation toward the bottom of the containers. The next layer (8 cm) consisted of coarse gravel, followed by a third layer (40 cm) of coarse sand. A fourth layer (25 cm) was filled with loess from a local site. The final layer (25 cm) was filled with a substrate mixture (two parts homemade compost, one part white peat substrate TS 1 (Klasmann-Deilmann GmbH, D-49744 Geeste), one part sand). Five liters of water per container (3 L in the early morning, 2 L at night) were supplied by a droplet irrigation system thrice a week; in total, 15 L water per week. Fertilization was applied as in the field one month after transplanting by 50 g/container Nitrophoska 15+15+15 (+2S) 15% Nitrogen, 15% $P_2O_5$, 15% $K_2O$, 2% S (EuroChem Agro GmbH, D-68165 Mannheim).

In the glasshouse, plants were grown in single pots (5.5 L volume), which were arranged in 10 rows of 11 plants, each with one plant of every genotype at randomized positions within the rows. Pots were filled with the same substrate mixture as used in the top layer of the container-based system in the PhenoSphere. Manual irrigation was applied daily to ensure continuous well-watered conditions. Plants were fertilized once a week with the irrigation in a concentration of 2‰ dissolved in the irrigation water. Before tassel emergence, Hakaphos blue (15% N (4% Nitrate and 11% Ammonium), 10% $P_2O_5$, 15% $K_2O$, 2% MgO) and after tassel emergence, Hakaphos red (8% N (3% Nitrate and 5% Ammonium), 12% $P_2O_5$, 24% $K_2O$, 4% MgO) was applied (COMPO EXPERT GmbH, D-48155 Münster).

After a 4-day acclimation phase in the PhenoSphere and the large glasshouse, plants were transplanted. Environmental conditions in the PhenoSphere were set to mimic the average weather of the same time of the years 2016/17/18 starting with the beginning of May and ending with the end of August. Hourly aggregated local weather data were derived from a standard Lambrecht weather station on the IPK grounds equipped with a 180° pyranometer 16130 0–2000 w/m² and a global range of 285–3000 nm (Lambrecht Meteo GmbH, D-37085 Göttingen). All sensor data from all environments can be found in Supplementary Data 1.

For every calendar week to be simulated, seven median days were calculated from the 21 template days (3 years × 7 days/week) from 2016/17/18. One representative sunny and one cloudy day with high and low day temperatures, respectively, was chosen from the daily medians for any given week. The normal day for each week was derived by averaging the hourly temperature of the seven median days. The seven days of a week were simulated from three normal days, two sunny and two cloudy days. Their sequence was randomized once at the beginning of the experiment and repeated every week (normal: Friday, Tuesday, Wednesday; cloudy: Sunday, Thursday; sunny: Saturday, Monday). Daily air humidity profiles in the field were rather uniform over the growth periods in all years. Therefore, air humidity in the PhenoSphere was set to 90% relative humidity during the night and to a gradual decline to 40% (on normal/sunny days) and 50% (on cloudy days) during the five central hours of the light period. Sudden changes in humidity values were avoided by programming ramps between the different levels. The illumination was controlled for normal, sunny, and cloudy days individually, and light intensities and fraction of day covered under clouds were predefined (Supplementary Tables 4–6).

On normal days, 5 h of cloud cover was simulated through light intensity fluctuations (intensity changes in a sinus curve). Sunny days included only 2 h of cloud cover simulation (fluctuating light), and cloudy days 8 h of cloud cover simulation (fluctuating light). Light intensities were highest on sunny days, only marginally lower on normal days (around 961 and 940 µmol m$^{-2}$ s$^{-1}$ on sunny and normal days, respectively) and low on cloudy days (319 µmol m$^{-2}$ s$^{-1}$). The length of the day was changed every week by 10 min earlier sunrise and 10 min later sunset with a reciprocal pattern after the 21st of June. Moderate wind movement was simulated by five big fans (PBT/4-630/32, Soler & Palau Deutschland GmbH, D-64293 Darmstadt) on each side of the PhenoSphere compartment (left and right perpendicular to the plant rows). Wind direction was changed every 2 h, and wind intensity oscillated between 10 and 20% of maximal rotation speed during night. Wind intensity during the days was split into ramping between 30 and 50% in the morning and in the late afternoon and into a constant 50% during midday. At 100% rotations, fans reached a maximum volumetric flow rate of 16,450 m³ h$^{-1}$, which translated into wind speeds of 6 m s$^{-1}$ at 6 m distance and gradually decreased to 2 m s$^{-1}$ at 28 m distance.

To simulate/recreate the weather characteristics of the growing season 2016 ('PhenoSphere 2016 sim'), the hourly recorded weather profiles from the field site at the IPK of just the year 2016 were used. For any recreated week, the temperature profile of the hourly averages of the 7 days of the week was deemed to define the normal day type of this week. The sunny and cloudy days of a week were modeled from individual days selected according to the same criteria as for the 'PhenoSphere avg' experiment. Instead of using the same number of the three representative days in reoccurring patterns, the frequencies and patterns were adjusted individually for each week. If a week, e.g., had more sunny days, they would have been represented more often in frequency, and they would have been repeated consecutively instead of distributing them randomly to better reflect reality as low- and high-pressure areas usually persist for more than one day.

In the GH, conditions were set to 25 °C/18 °C temperature and 80%/60% relative humidity at day and night, respectively, with 16 h/8 h day-night cycle. Assimilation light was switched on when the sensor on the roof dropped below 60 klx between 7 am and 11 pm, and glass panels were partially shaded to maintain a stable light intensity of around 250 µmol m$^{-2}$ s$^{-1}$ in the glasshouse.

The homogeneity test experiment in the PhenoSphere was run with the genotypes PHT77 and N22 in every container to estimate the effects of container position and potential effects of preculture of the 'PhenoSphere avg' experiment in the containers as the soil was

reused. The containers were randomly sorted into three groups. For 25 containers, the soil from the preceding experiment was reused unchanged. For the other 25 containers, the top soil layer was exchanged with fresh substrate mixture, and for the remaining five containers, all soil layers were renewed and the containers were completely refilled. In each container, the two genotypes were randomly distributed between side 1 and 2. The environment simulation of the 'PhenoSphere avg' experiment running the averaged weather conditions of the years 2016/17/18 was reused in this experiment. As in the 'PhenoSphere avg' experiment, fertilizer was applied 1 month after transplanting with 50 g/container Nitrophoska 15+15+15 (+2S) 15% Nitrogen, 15% $P_2O_5$, 15% $K_2O$, 2% S (EuroChem Agro GmbH, D-68165 Mannheim).

## Phenotyping

Phenotypic traits of maize plants were measured manually/visually in regular intervals throughout the cultivation phase of every cultivation scenario (Supplementary Data 2). Plant Height was defined as the distance from the soil to the highest point of a plant. The vegetative stage (V-stage) was determined as the number of leaves that are mature and express a visible leaf collar[24,25]. The trait growth stage complementarily counts the total number of leaves visible at any time point. The lower leaves were manually marked with metal rings before they deteriorated to ensure precise counting. The final vegetative development stage (VT) was defined as the ratio between the growth stage and V-stage approaching 1. The occurrence of tassels was visually scored during every measurement as either present or absent. After the VT-stage, plants were kept in their respective environment until the ears were mature. The ears per plant were counted, harvested and total grain number and total grain weight per plants was measured. Measurement raw data for all traits and every environment are provided with this paper as well as the sensor data logged.

Vapor pressure deficit (VPD) was calculated from the relative air humidity and the air temperature with the equation from Buck 1981[26,27] with updated improved empirical values for aw = 6.1121, bw = 18.678, cw = 257.14, and dw = 234.5, taken from Buck Research Instruments L.L.C. operating manual for the CR-1A Hygrometer (http://www.hygrometers.com/wp-content/uploads/CR-1A-users-manual-2009-12.pdf).

## Best linear unbiased estimators of the phenotypic traits

Best linear unbiased estimators (BLUEs) were obtained for the traits plant height, growth stage, and vegetative stage. Linear mixed models (LMM) were fit with the *lme4* package version 1.1.27.1[28]. The random-effect structures of all LMM, fitted to estimate BLUEs, were kept maximal if justified by the experimental design[29].

$$
\begin{aligned}
\text{trait}_i &\sim N(\alpha_{j[i],k[i],l[i],m[i],n[i],o[i],p[i]} + \beta(\text{Genotype}), \sigma^2) \\
\alpha_j &\sim N(\mu_{\alpha_j}, \sigma^2_{\alpha_j}), \text{for Replicate : Row } j = 1, \ldots, J \\
\alpha_k &\sim N(\mu_{\alpha_k}, \sigma^2_{\alpha_k}), \text{for Replicate : Position } k = 1, \ldots, K \\
\alpha_l &\sim N(\mu_{\alpha_l}, \sigma^2_{\alpha_l}), \text{for Replicate : Column } l = 1, \ldots, L \\
\alpha_m &\sim N(\mu_{\alpha_m}, \sigma^2_{\alpha_m}), \text{for Column } m = 1, \ldots, M \\
\alpha_n &\sim N(\mu_{\alpha_n}, \sigma^2_{\alpha_n}), \text{for Row } n = 1, \ldots, N \\
\alpha_o &\sim N(\mu_{\alpha_o}, \sigma^2_{\alpha_o}), \text{for Position } o = 1, \ldots, O \\
\alpha_p &\sim N(\mu_{\alpha_p}, \sigma^2_{\alpha_p}), \text{for Replicate } p = 1, \ldots, P
\end{aligned} \tag{1}
$$

The model for the 'PhenoSphere avg' and 'PhenoSphere 2016 sim' experiments is defined in (1). The 11 genotypes were fitted as fixed effects and the effects of replicate ($\alpha_p$), of the position of a plant in the container ($\alpha_o$), of the row ($\alpha_n$) and column ($\alpha_m$) position of a container, the interaction between replicate and row ($\alpha_j$), the interaction between replicate and position ($\alpha_k$), and the interaction between replicate and

column ($\alpha_l$) as random effect with the error term $\varepsilon \sim N(0, \sigma^2_\varepsilon)$. BLUEs were estimated individually for each date of measurement from the genotypes fitted as fixed effects.

The model for the glasshouse experiment was defined in (2):

$$
\begin{aligned}
\text{trait}_i &\sim N(\alpha_{j[i],k[i]} + \beta(\text{Genotype}), \sigma^2) \\
\alpha_j &\sim N(\mu_{\alpha_j}, \sigma^2_{\alpha_j}), \text{for Replicate } j = 1, \ldots, J \\
\alpha_k &\sim N(\mu_{\alpha_k}, \sigma^2_{\alpha_k}), \text{for Row } k = 1, \ldots, K
\end{aligned} \tag{2}
$$

The genotypes were fitted as fixed effects again and replicate ($\alpha_j$) and row of the pots in the greenhouse ($\alpha_k$) as random effects with the error term $\varepsilon \sim N(0, \sigma^2_\varepsilon)$.

The model for the field experiments in 2016/17/18/19 is defined in (3):

$$
\begin{aligned}
\text{trait}_i &\sim N(\alpha_{j[i],k[i],l[i]} + \beta(\text{Genotype}), \sigma^2) \\
\alpha_j &\sim N(\mu_{\alpha_j}, \sigma^2_{\alpha_j}), \text{for Replicate : Position } j = 1, \ldots, J \\
\alpha_k &\sim N(\mu_{\alpha_k}, \sigma^2_{\alpha_k}), \text{for Position } k = 1, \ldots, K \\
\alpha_l &\sim N(\mu_{\alpha_l}, \sigma^2_{\alpha_l}), \text{for Replicate } l = 1, \ldots, L
\end{aligned} \tag{3}
$$

Genotypes were fitted as fixed effects and replicate ($\alpha_l$), position in the field plot ($\alpha_k$), and its interaction ($\alpha_j$) as random effects with the error term $\varepsilon \sim N(0, \sigma^2_\varepsilon)$.

## Nonlinear mixed-effect models

Differences between cultivation scenarios (field, glasshouse, and PhenoSphere cultivations) were estimated with nonlinear mixed-effect models using the *nlme* package version 3.1–152 in R version 4.1.0[30,31]. Model fitting followed suggestions from other researchers[32,33]. Prior to fitting nonlinear mixed-effect models, data were grouped by crossing the cultivation and the genotype factors into 77 groups.

A three-parameter simple logistic growth model $y(x) = \frac{\phi_1}{1 + e^{\left(\frac{\phi_2 - x}{\phi_3}\right)}}$, *SSlogis*[30,32] was fit for the parameters $\phi_1$ = "Asym", $\phi_2$ = "xmid", and $\phi_3$ = "scal" on the BLUEs of plant height as a complete random effects model assuming zero autocorrelation. Intercepts of the random effects model were used as starting values for fixed effect estimation to update the model to a mixed-effect model with cultivations as fixed effects and genotypes as random effects, which was justified by a lower Akaike information criterion (delta AIC) of 82.6. Variances were allowed to differ for each cultivation scenario with the constant variance function (*varIdent*), which improved the model fit by delta AIC 15.7. Autocorrelation between the measurements during the timeseries was accounted for by an autoregressive-moving average (ARMA) correlation structure of order ($p = 2$, $q = 2$) for a global intercept found by the *auto.arima* function from the *forecast* package version 8.15[34], justified by a delta AIC of 1654 to the mixed-effect model. The ARMA ($p = 2$, $q = 2$) effectively accounted for the first 10 lags of the model (Box-Ljung test, X-squared = 4.5, df = 10, p-value = 0.9). The model was updated by maximum likelihood. The standard deviation for the random effect of the parameter "scal" was found to be close to zero and thus the random effect for "scal" was removed, increasing model fit with a delta AIC 2. The model for plant height was then refit with restricted maximum likelihood, which further decreased the AIC by 80.8. The model fit to the BLUEs of plant height is shown in Supplementary Fig. 6. From the final model, individual "Asym" and "xmid" values were extracted from the random effects for each genotype and "scal" was the same for all genotypes, as it was best fitted as a fixed effect. Temporal development of plant height for each individual genotype, calculated from this data, was then plotted in Supplementary Fig. 12.

To cluster the plant height of each genotype in each environment at 68 DAS, the average maximum growth speed for the four field

environments and the 'PhenoSphere 2016 sim', the modeled logistic growth function *SSlogis* was solved individually for an "xmid" value of 68 DAS. The hierarchical clustering was done with the *heatmap.2* function of the *gplots* package version 3.1.1[35] calculating a Manhattan distance matrix and performing a complete-linkage clustering scaled for rows to have a mean zero and standard deviation of one (Supplementary Fig. 7).

For the trait vegetative stage (V-stage), a random effects model was fit with the *SSlogis* function, assuming zero correlation between parameters. Including fixed effects into a mixed-effect model, fit with the intercepts of the random-effects model, improved model fitting (delta AIC 30.2). Variance modeling allowing different standard deviations per cultivation further improved the model (delta AIC 2.8). An autocorrelation structure of ARMA ($p = 2$, $q = 2$) for time as primary covariate grouped by crossed cultivation and genotype, determined by *auto.arima*, accounted for time series dependence (Box-Ljung test, X-squared = 5.1, df = 10, $p$-value = 0.89) justified by a delta AIC of 2711.9. The random effect of the "scal" parameter was removed (delta AIC 2) after maximum likelihood estimation. The final model was estimated with REML (delta AIC 50.8). The model fit to the BLUEs of V-stages is shown in Supplementary Fig. 8. As for plant height "Asym" and "xmid" from random effects and one "scal" from the fixed effect was used to calculate best fit model to leaf maturation of each genotype (Supplementary Fig. 13).

The data from counting all leaves per plants (growth stage) was also fit with the *SSlogis* function assuming zero correlation between parameters. Updating the model with the intercepts from the random-effects model as starting values into the mixed-effect model improved its fitting (delta AIC 36.5). The autocorrelation structure most suitable for the growth stage data for time as primary covariate grouped by crossed cultivation and genotype was found to be represented by an ARMA ($p = 4$, $q = 2$) by the *auto.arima* function (Box-Ljung test, X-squared = 1.42, df = 10, $p$-value = 0.99). This drastically improved model fitting (delta AIC 2368.9). After estimation of the model with maximum likelihood standard deviations of the random parameters for "xmid" and "scal" were found to approach zero and were removed (delta AIC 3). The final model was estimated by restricted maximum likelihood (delta AIC 52). The best fit model to the BLUEs of the growth stages can be seen in Supplementary Fig. 10. To plot individual trajectory of growth stage development for each genotype (Supplementary Fig. 14), "Asym" values from the random effects were determined genotype-wise and "xmid" and "scal" were taken model-wide from the fixed effects.

Bonferroni-corrected 95% confidence intervals and $p$-values were estimated for differences between the environments for all traits (Supplementary Data 3) for the parameters "asym", "xmid", and "scal" with the *emmeans* package version 1.6.2–1[36].

The appearance of tassels is a binary characteristic in that tassels are either present or not. Tasseling was measured for each replicate of each genotype but aggregated as total percentages of tasseling plants over all genotypes for each cultivation at each time point. A fixed-effect nonlinear regression model was fit with the *nls* function for the logistic growth function formula from the *SSlogis* model[32]. Measurement time points and the cultivations were treated as fixed effects. Residuals and fitted values of the fixed-effect model were analyzed in a two-way ANOVA with time and cultivation as factors. Neither factor significantly affected the residuals of the model, but both factors had a significant effect on the fitted values (factor "time", df = 83, $F = 23.5$, $p$-value = < 0.001; factor "cultivation", df = 6, $F = 17.0$, $p$-value = < 0.001). The model fit to the percentages of plants with tassel is plotted in Supplementary Fig. 16.

The repeatability for the population of 11 genotypes over all cultivation scenario was approximated with a mixed-effects model (4). To reduce heteroscedasticity, the trait plant height was square root transformed. Model fit was optimized with the *AICtab* function from the *bbmle* package version 1.0.24[37].

$$\begin{aligned}
\text{Plant\_Height}_i &\sim N(\alpha_{j[i],k[i],l[i]} + \beta_1(\text{Experiment}) + \beta_2(\text{Experiment time}), \sigma^2) \\
\alpha_j &\sim N(\mu_{\alpha_j}, \sigma^2_{\alpha_j}), \text{for Genotype : Experiment time } j = 1, \ldots, J \\
\alpha_k &\sim N(\mu_{\alpha_k}, \sigma^2_{\alpha_k}), \text{for Genotype : Experiment } k = 1, \ldots, K \\
\alpha_l &\sim N(\mu_{\alpha_l}, \sigma^2_{\alpha_l}), \text{for Genotype } l = 1, \ldots, L
\end{aligned}$$

(4)

The effects of the experiments and the experiment time, which is the date of measurement in each experiment were fit (4) as fixed effects. All other effects were fit (4) as random for genotype ($\alpha_l$), for the interaction between genotype and experiment time ($\alpha_j$), and for the interaction between genotype and experiment ($\alpha_k$). The repeatability for the genotype ($R_g = 0.55$) was calculated with $R_{\text{genotype}} = \frac{\sigma^2_{\alpha_l}}{\sigma^2_{\alpha_l} + \sigma^2_\varepsilon}$; $\sigma^2_{\alpha_l}$ being the between-group variance for the genotypes and $\sigma^2_\varepsilon$ the residual variance from the error term $\varepsilon \sim N(0, \sigma^2_\varepsilon)$. The marginal coefficient of determination ($r^2_m = 0.866$) and the conditional coefficient of determination ($r^2_c = 0.961$) of the model (4) were estimated with the *MuMIn* package version 1.43.17[38].

## Variance decomposition of positioning effects in the PhenoSphere homogeneity test experiment

To determine the influence of positioning effects in the PhenoSphere on variance composition, repeatability was approximated from a linear mixed-effect model on plant height with the *rptR* package version 0.9.22[21]. Plant height data from the homogeneity test experiment in 2019 were square root transformed to reduce heteroscedasticity.

$$\begin{aligned}
\text{Plant.Height}_i &\sim N(\alpha_{j[i],k[i],l[i],m[i],n[i]} + \beta(\text{Genotype}), \sigma^2) \\
\alpha_j &\sim N(\mu_{\alpha_j}, \sigma^2_{\alpha_j}), \text{for Preculture } j = 1, \ldots, J \\
\alpha_k &\sim N(\mu_{\alpha_k}, \sigma^2_{\alpha_k}), \text{for Column } k = 1, \ldots, K \\
\alpha_l &\sim N(\mu_{\alpha_l}, \sigma^2_{\alpha_l}), \text{for Row } l = 1, \ldots, L \\
\alpha_m &\sim N(\mu_{\alpha_m}, \sigma^2_{\alpha_m}), \text{for Replicate } m = 1, \ldots, M \\
\alpha_n &\sim N(\mu_{\alpha_n}, \sigma^2_{\alpha_n}), \text{for Soil } n = 1, \ldots, N
\end{aligned}$$

(5)

The genotypes were fit (5) as fixed effects, and the effects of preculture ($\alpha_j$), column ($\alpha_k$), row ($\alpha_l$), replicate ($\alpha_m$), and soil treatment ($\alpha_n$) were fit as random effects with the error term $\varepsilon \sim N(0, \sigma^2_\varepsilon)$. Variance composition was individually determined at each measurement date (Supplementary Table 3) and plotted (Supplementary Fig. 17). *rptR* was specified to determine the agreement repeatability[39] with confidence intervals for 1000 bootstrapping cycles.

## Thermal time scaling

Cumulative thermal time (*tt*) was calculated in (6) for every hour of a day and for every day of a growing season on the fields, in the PhenoSphere, and in the glasshouse[20]. In occasions with missing data, data were linearly imputed to fill the gaps.

$$tt_n = (tt_{n-1}) + \frac{1}{24} * \frac{T * e^{\left(\frac{-\Delta H_A}{R*T}\right)}}{1 + \left[e^{\left(\frac{-\Delta H_A}{R*T_{20}}\right)}\right]^{\alpha\left(1 - \frac{T}{T_o}\right)}} \Bigg/ \frac{T_{20} * e^{\left(\frac{-\Delta H_A}{R*T_{20}}\right)}}{1 + \left[e^{\left(\frac{-\Delta H_A}{R*T_{20}}\right)}\right]^{\alpha\left(1 - \frac{T_{20}}{T_o}\right)}}$$

(6)

In (6), $n$ is the hour beginning with 1, $T$ is the temperature in Kelvin, $-\Delta H_A = -73900 J*mol^{-1}$ is the enthalpy of activation specific for maize, $T_{20} = 293$ K is the temperature at 20 °C in Kelvin, $T_o = 306.4$ K is the maize specific maximum temperature in Kelvin,

$R = 8.314 J*mol^{-1}*K^{-1}$ is the gas constant, and $\alpha = 3.5$ is a unitless parameter[20]. Differences between plant height in cultivation scenarios (field, glasshouse, and PhenoSphere cultivations) for thermal time days were estimated with nonlinear mixed-effect models like described above. A four-parameter simple logistic growth model $y(x) = \phi_1 + \frac{\phi_2 - \phi_1}{1 + e^{\left(\frac{\phi_3 - x}{\phi_4}\right)}}$ $SSfpl$[30,32] was fit for the parameters $\phi_1 = $ "A" horizontal asymptote on the left side, $\phi_2 = $ "B" horizontal asymptote on the right side, $\phi_3 = $ "xmid", and $\phi_4 = $ "scal" on the BLUEs of plant height as a complete random effects model assuming zero autocorrelation. Bonferroni-corrected 95% confidence intervals and $p$-values were estimated for differences between the environments for all model parameters (Supplementary Data 3).

## Yield components
The yield parameters ear count, grain number, and total grain weight were analyzed by calculating the best linear unbiased predictors (BLUPs) using the same Eqs. (1, 2, and 3) for the respective environments but treating the genotype as a random effect. BLUPs were extracted with the *coef* functions in R[31]. From a simple mixed model with cultivation as fixed effect and genotypes as random effects, the estimated means (*emmeans::emmeans*) for each cultivation were plotted, and pairwise Tukey-adjusted $p$-values (*emmeans::pwpm*) were computed[36]. Confidence intervals were determined via the *confint* function for the 0.95 level on the basis of the genotypes fitted as random effects.

## Reporting summary
Further information on research design is available in the Nature Portfolio Reporting Summary linked to this article.

# Data availability
The authors declare that sensor and phenotypic data from each environment generated in this study are provided in Supplementary Data 1 and 2, respectively. The processed data are provided within the Supplementary Code 1 folder. Six maize accession (ZEA 132 https://doi.org/10.25642/IPK/GBIS/33630, ZEA 324 https://doi.org/10.25642/IPK/GBIS/33799, ZEA 332 https://doi.org/10.25642/IPK/GBIS/33807, ZEA 3660 https://doi.org/10.25642/IPK/GBIS/234225, ZEA 399 https://doi.org/10.25642/IPK/GBIS/70927, ZEA 851 https://doi.org/10.25642/IPK/GBIS/70928) were sourced from the IPK Gene Bank (https://gbis-ipk-gatersleben.de/gbis2i/).

# Code availability
The authors declare that all code supporting the findings of this study is available within the folder Supplementary Code 1, which contains R scripts and the necessary source files to reproduce every figure and statistics presented in this study.

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

## Acknowledgements

We thank Gunda Wehrstedt, Iris Fischer, Marion Michaelis, Alexandra Rech, Andrea Apelt, Sibille Bettermann, Monika Gottowik, and Beatrice Knüpfer for their excellent technical assistance. The authors acknowledge financial support from the German Plant Phenotyping Network (DPPN), which is funded by the German Federal Ministry of Education and Research (BMBF) (project identification number: 031A053) and from the German Federal State of Saxony-Anhalt. Costs for open access publishing were partially funded by the Deutsche Forschungsgemeinschaft (DFG, German Research Foundation, grant 491250510).

## Author contributions

M.H., A.J., and T.A. conceived the experimental idea and designed the experiments. T.A., A.J., M.H. and D.K. developed the PhenoSphere Infrastructure. A.J. provided the first batch of maize seeds. M.H., A.J. and D.K. performed the phenotyping experiments. M.H. processed and analyzed the data and performed the statistics. M.H. and T.A. wrote the manuscript with input from all authors.

## Funding

## Competing interests

The authors declare no competing interests.
