## [Peer Review File · Nature Communications]

Reviewers' Comments:

Reviewer #1:

Remarks to the Author:

The topic addressed by the study is fully relevant in terms of phenotyping and suited for Nature Communications and the paper is in general well written.

However, the results achieved are rather limited considering the resources deployed.

First of all, genotypic differences are not addressed but just pooled data is used.

Second, getting growth patterns (growth speed, plant height, leaf maturation) and phenology (tassel occurrence) comparable to those under field conditions is important but this is just part of the issue. If the study claims that PhenoSphere mimics field conditions, then total biomass (even if from the fully mature crops) or even grain yield have to be included in the study to show values are comparable to these achieved under field conditions. Otherwise I don't see a dramatic advantage from the already existing growing systems.

If (as stated in the Abstract), the PhenoSphere, is "a plant cultivation infrastructure designed to simulate field-like environments in a reproducible manner.", then achieved grain yield, biomass and other key agronomical yield components should resemble field conditions. The same concept about the capabilities of PhenoSphere" is stressed through the paper . For example Line 268-269: "In the PhenoSphere, a typical field-like environment can be simulated, indeed."

Lines 281-287. Again if simulating the relevant agronomic traits is the target, then plant height and flowering time are not the most relevant. Crop models aim to simulate and predict biomass and yield. so it is necessary to prove the performance of the PhenoSphere with regard such a traits.

From Fig. 3 it appears that maize in the greenhouse was grown under really low light conditions

Extended data Fig. 1. The field layout (particularly the distance between plots) is quite unusual and doesn't resemble a maize trial.

Minor things

Line 168-169. Rewrite the sentence

Line 337: Complete scientific name for maize

Reviewer #2:

Remarks to the Author:

This MS presents experiments carried out in a novel facility, PhenoSphere, which is an impressive large growth chamber. Results show that temperatures in three field experiments can be reproduced in this facility, and that plants have similar phenology and height as in field experiments. An experiment in a greenhouse is also presented, whose role in the reasoning is not fully clear. I have six major points and some minor points

1. I would have expected that the facility is fully presented in the 'method' section, in particular its geometry. This is only shown via two images. It is suggested that the facility is high in relation to plants, but this is not precisely indicated except if I am mistaken. Similarly, it would be useful to state the area of the facility, how many plants can be held simultaneously, the way in which light is provided, maximum light intensity at plant level etc.

2. Climatic conditions are presented via temperature only. This is not sufficient for convincing the reader that environmental conditions resemble those in fields. Indeed, temperature can be controlled adequately in most phenotyping facilities, so this is not original per se. More important would be to present light intensity at plant level, its vertical distribution, the light spectrum and the vapour pressure deficit in the facility. I understand from the 'method' section that this data was collected.

- Light intensities are lower in most greenhouse or growth chamber compared with those during a sunny day in the field (typically 200 – 500 and 1000 $\mu\text{mol m}^{-2} \text{s}^{-1}$, respectively, vs 1500 to 2000 $\mu\text{mol m}^{-2} \text{s}^{-1}$ outside). It would make a real difference if PhenoSphere could provide such light intensity. Similarly, the light distribution in the canopy differs between growth chambers and field because lamps are close to plants in growth chambers. It is suggested that this is not the case in PhenoSphere, potentially solving this problem. This would be a large advantage.

- Light spectrum is another issue in greenhouses and growth chambers, it is very difficult to reproduce the spectrum of solar light. It is stated in the MS that plants tend to be higher in PhenoSphere than in the field, suggesting that the spectrum is different in PhenoSphere compared with outside.

- Vapour pressure deficit is much lower in most greenhouses and growth chambers compared with the field (difficult to get higher VPDs than 1.5 – 2 kPa, vs 3.5 kPa during sunny days outside). Is this problem solved in PhenoSphere?

It would be necessary to consider these variables in the comparison of environmental conditions in field and PhenoSphere, so the reader can be convinced that PhenoSphere can adequately represent fields. I would expect that PhenoSphere will not match all criteria, but it would be important to know on which criteria PhenoSphere can represent the field, and to discuss the potential pitfalls for others (see point 6 about the discussion).

3. I was not convinced by the statistical method for comparing temperatures in the field and in PhenoSphere. The issue is not to know if corresponding temperatures are independent or not (this is what is tested via R2 and pvalue), but if any bias exists between PhenoSphere and the field. I suggest that daily temperatures are compared between environments via RMSE, which calculates the error when considering that PhenoSphere temperatures represent those in the field. I also suggest more integrated indicators, (i) Thermal time, to test if errors accumulate with time, potentially causing a bias in the prediction of developmental stages or, alternatively, if they compensate over time (ii) the number of days with afternoon temperature higher than a threshold, typically 30°C for maize, and with night temperatures lower than a threshold, typically 14°C for maize, to test if extreme events occur with similar frequencies.

I would suggest that the same methods are used to compare light and VPD between PhenoSphere and field.

4. I was disappointed by the phenotypic variables taken into account for comparing plants in PhenoSphere and in the field. Time to anthesis and plant height are rather crude variables in a phenomic study, for such a sophisticated facility. Furthermore, flowering time is usually well predicted via thermal time in all environments (e.g. multiple experiments in growth chamber, field and greenhouse). It is therefore expected that they match here between PhenoSphere and the field, this is not original. It would be interesting to know to what extent the relation between thermal time and time to anthesis is common to all environments, I expect it will be the case. Similarly, the difference in the date of maximum growth, which differs between experiments, may be accounted for by differences in thermal time. It would be worth testing this rather than stating that these dates either differ or are similar between experiments.

5. Experiments were carried out with 11 maize lines, but only mean values for all genotypes are presented. It would be important to test to what extent the ranking of genotypes differs or not between environments.

6. The discussion repeats that PhenoSphere is the missing link between field and controlled conditions, without further arguments than in the introduction. I would expect a more thorough discussion: what can be done in PhenoSphere that cannot be done in a field equipped with sensors? or an equipped greenhouse? Pros and cons should be discussed, e.g. in line with the table proposed by Poorter et al in their paper in *New Phytologist* 2016 212,838

Minor points

Line 97 and below. The term 'modelled' is not appropriate, there is no model here. 'Programmed' would probably be better.

Line 98 and below. Similarly, the word 'variables' should be preferred to 'parameters' for environmental variables. Parameter has a strict meaning in modelling, namely a characteristic that is specific to an object and does not change with time, e.g. albedo or parameters of a soil water release curve.

Line 161 Differences in date between environment may be accounted for by differences in temperature (they probably are). This can be easily tested via the use of thermal time in all environments, for answering the question 'is this due to bias in temperature, or to a more physiological reason'.

Line 183. I do not understand which variance is meant here

Line 201. These values of heritability are not very different from those in a well-managed field experiment?

Reviewer #3:
None

Revision

Point-by-point response to the reviewers' comments
(line numbers refer to the revised manuscript document with tracked changes)

REVIEWER COMMENTS and RESPONSES

Reviewer #1 (Remarks to the Author):

The topic addressed by the study is fully relevant in terms of phenotyping and suited for Nature Communications and the paper is in general well written.

However, the results achieved are rather limited considering the resources deployed.

First of all, genotypic differences are not addressed but just pooled data is used.

Response:

Thank you for your very positive overall judgement of our work and for the hint on the genotypic differences. We failed to include the data of the individual genotypes in the results section and referred to the corresponding supplementary figures only in the discussion. We changed that and pointed the reader to the GxE interaction which is present in each environment and also between the field 2016 and the 'PhenoSphere 2016 sim' experiment. See l. 214 – 217 of the revised manuscript.

In the supplementary figures 12 – 15 we show the genotype-wise individual growth trajectory of every genotype in every environment for every trait tested, except for 'plants with tassels' due to the nature of the data.

Second, getting growth patterns (growth speed, plant height, leaf maturation) and phenology (tassel occurrence) comparable to those under field conditions is important but this is just part of the issue. If the study claims that PhenoSphere mimics field conditions, then total biomass (even if from the fully mature crops) or even grain yield have to be included in the study to show values are comparable to these achieved under field conditions. Otherwise I don't see a dramatic advantage from the already existing growing systems.

Response:

We fully agree with the reviewer that comparisons of (grain) yields achieved in field trials and in the PhenoSphere are highly important. After the achievements made in the experiments presented in this manuscript, we therefore initiated further investigations specifically designed for this purpose and look forward to publish these in due course. The goals of the present study were to test, optimize, and validate the ability to elicit growth habits and developmental programs of (crop) plants very similar those under field conditions by exposing them to well-designed growth conditions in the PhenoSphere. In order to be able to learn how to design and optimize the environmental control programs to best mimic field conditions and to demonstrate this, we deemed it necessary to monitor the responses of a suitable population of plants continuously during all stages of their life cycle. We considered it essential to gain phenotype data (related to growth and developmental stages) at

numerous time points and at regular intervals throughout the entire cultivation season (rather than to measure single endpoint values of a highly complex trait such as seed yield) in order to be able to detect relevant effects of the way the environment simulation was carried out (condition x time point). For practical reasons (e.g. length of the cultivation period), we decided to perform these series of experiments using a plant species with an annual life cycle and with high importance as a crop in Europe and worldwide, such as maize. In order to avoid a potential bias related to a certain phenology type (potentially with specific responses to environmental cues at certain periods), we furthermore decided that a population of lines with high diversity in terms of growth and development has to be used for this purpose. This led to the selection of the 11 maize inbred lines, which cover a very wide range of growth habits and phenology (e.g. flowering time). This diverse population enabled the continuous monitoring of a very broad range of growth and developmental responses during the entire cultivation period, which turned out to be very important considering the observations made during the first experiment in the PhenoSphere simulating average conditions of the years 2016 – 2018. However, the selected population and the devised experimental design precluded the collection of meaningful yield data, in particular for an outcrossing species such as maize. For conclusive yield trials, a very different population of genotypes should be used: The genotypes should vary substantially with respect to yield, but they need to be of the very same phenology class with a very narrow range of flowering times. Furthermore, a different experiment design is required for yield trials (with larger plots resembling better the plant stands in the field and ensuring appropriate pollination of the assessed plants), both for the outdoors and for the indoors trials.

Corresponding remarks were added to the discussion (see l. 244 – 251 & 311 – 313).

We consider the advantages of the PhenoSphere (over standard climatized glass houses) and the advances in the programmed simulation of appropriate dynamic environmental conditions (over averaged or simple scenarios) very substantial and important. As mentioned above, experiment series have been initiated (currently using selfing species) in which (seed) yield data are collected and with which we'd like to address the highly relevant question of comparing (grain) yields achieved in field trials and in the PhenoSphere.

If (as stated in the Abstract), the PhenoSphere, is "a plant cultivation infrastructure designed to simulate field-like environments in a reproducible manner.", then achieved grain yield, biomass and other key agronomical yield components should resemble field conditions. The same concept about the capabilities of PhenoSphere" is stressed through the paper . For example Line 268-269: "In the PhenoSphere, a typical field-like environment can be simulated, indeed."

Response:

As outlined above, we agree with the reviewer that data on yield, biomass and key agronomical yield components need to be generated in order to consider the PhenoSphere as an "indoor field", in which yield of crops achieved in particular field-like scenarios can be precisely assessed. As outlined above, different sets of genotypes need to be used for this and the experimental design has to be adjusted for this purpose.

The manuscript text was edited to clarify the goals and the limits of the present study and to point out the need of yield trials as part of the further assessment of the capabilities of the PhenoSphere and its utility (see: l. 311 – 313).

Lines 281-287. Again if simulating the relevant agronomic traits is the target, then plant height and

flowering time are not the most relevant. Crop models aim to simulate and predict biomass and yield. so it is necessary to prove the performance of the PhenoSphere with regard such a traits.

Response:

We agree with the reviewer: In fact, we see a major future use of the PhenoSphere as the prime growth and phenotyping facility to experimentally test predictions from crop models, ideally combined / enriched with metabolic models, that are constrained with information from systems biological analyses and with certain environmental factors: The environmental scenarios considered in the models may be run in the PhenoSphere and the performance of genotypes predicted to produce high/low biomass and/or yield may be tested experimentally. As mentioned before, in ongoing and future projects, we intend to collect (seed) yield data in the PhenoSphere and compare them with results of field trials in order to prove the outlined use of the PhenoSphere.

From Fig. 3 it appears that maize in the greenhouse was grown under really low light conditions

Response:

Indeed, as stated in line 524 – 526, light intensity in the glasshouse was maintained at $250 \mu\text{mol m}^{-2} \text{s}^{-1}$ with an outside glasshouse shading system and the indoor assimilation light sources. This relatively low light intensity is a compromise between using highly variable natural light and constant artificial illumination used to achieve reproducible conditions in the climatized glasshouse.

Extended data Fig. 1. The field layout (particularly the distance between plots) is quite unusual and doesn't resemble a maize trial.

Response:

Indeed, the chosen field design did not resemble that of a usual maize yield trial. The design with well-spaced double row plots was used for several reasons: With the spacing potential impacts of large/tall genotypes on neighboring small genotypes were avoided. It furthermore enabled frequent access to the plants and efficient manual phenotyping for the parameters of interest. It could readily be transferred to the container-system used in the PhenoSphere.

Minor things

Line 168-169. Rewrite the sentence

Response:

We modified the sentence to better describe what we have done (see l. 186 – 190).

Line 337: Complete scientific name for maize

Response:

We checked and corrected every entry of *Zea mays* in the manuscript.

Reviewer #2 (Remarks to the Author):

This MS presents experiments carried out in a novel facility, PhenoSphere, which is an impressive large growth chamber. Results show that temperatures in three field experiments can be reproduced in this facility, and that plants have similar phenology and height as in field experiments. An experiment in a greenhouse is also presented, whose role in the reasoning is not fully clear. I have six major points and some minor points

1. I would have expected that the facility is fully presented in the 'method' section, in particular its geometry. This is only shown via two images. It is suggested that the facility is high in relation to plants, but this is not precisely indicated except if I am mistaken. Similarly, it would be useful to state the area of the facility, how many plants can be held simultaneously, the way in which light is provided, maximum light intensity at plant level etc.

Response:

We added a full chapter in the M&M with comprehensive information of the technical specifications of the PhenoSphere and included additional Supplementary Figure 18 a – c showing its geometry/architecture. Furthermore, the light intensity values and their (horizontal) spatial variation at 3 vertical heights were included and are shown as an additional Supplementary Figure 20.

2. Climatic conditions are presented via temperature only. This is not sufficient for convincing the reader that environmental conditions resemble those in fields. Indeed, temperature can be controlled adequately in most phenotyping facilities, so this is not original per se. More important would be to present light intensity at plant level, its vertical distribution, the light spectrum and the vapour pressure deficit in the facility. I understand from the 'method' section that this data was collected. - Light intensities are lower in most greenhouse or growth chamber compared with those during a sunny day in the field (typically 200 – 500 and 1000 $\mu\text{mol m}^{-2} \text{s}^{-1}$, respectively, vs 1500 to 2000 $\mu\text{mol m}^{-2} \text{s}^{-1}$ outside). It would make a real difference if PhenoSphere could provide such light intensity. Similarly, the light distribution in the canopy differs between growth chambers and field because lamps are close to plants in growth chambers. It is suggested that this is not the case in PhenoSphere, potentially solving this problem. This would be a large advantage.

Response:

As mentioned above, in the additional M&M chapter and the additional Supplementary Figure, detailed information of the light intensity values and their spatial variation is given. At 20 cm, 120 cm, and 220 cm above soil level averages over the cultivation area of 1277, 1348, and 1436 $\mu\text{mol m}^{-2} \text{s}^{-1}$, PAR are achieved when all light sources are at 100%.

- Light spectrum is another issue in greenhouses and growth chambers, it is very difficult to reproduce the spectrum of solar light. It is stated in the MS that plants tend to be higher in PhenoSphere than in the field, suggesting that the spectrum is differs in PhenoSphere compared with outside.

Response:

In the additional M&M chapter on the technical specifications of the PhenoSphere, the light sources are given and another Supplementary Figure 19 is added, in which the spectra of all light sources individually and in combination and of the three model days, sunny, normal, and cloudy are shown.

- Vapour pressure deficit is much lower in most greenhouses and growth chambers compared with

the field (difficult to get higher VPDs than 1.5 – 2 kPa, vs 3.5 kPa during sunny days outside). Is this problem solved in PhenoSphere?

It would be necessary to consider these variables in the comparison of environmental conditions in field and PhenoSphere, so the reader can be convinced that PhenoSphere can adequately represent fields. I would expect that PhenoSphere will not match all criteria, but it would be important to know on which criteria PhenoSphere can represent the field, and to discuss the potential pitfalls for others (see point 6 about the discussion).

Response:

We are grateful for the suggestion to include information on the maximum VPD achieved in the PhenoSphere. The max. air temperature, rel. humidity, and VPD values achieved in the PhenoSphere upon 100% intensity of all light sources were included (see l. 391 – 394): 47.5 °C at 35 % relative humidity (VPD 6.94 kPa) and 37.5°C at 7.5 % relative humidity (VPD 5.99 kPa).

Furthermore, we calculated the hourly VPDs for the experiments in all seven environments and calculated the correlations among the environments and the RMSEs vs. the field season 2016 (see l. 135 – 138 and comments below). Highest correlation and lowest RMSE occurred between field 2016 and its simulation 'PhenoSphere 2016 sim'.

3. I was not convinced by the statistical method for comparing temperatures in the field and in PhenoSphere. The issue is not to know if corresponding temperatures are independent or not (this is what is tested via R2 and pvalue), but if any bias exist between PhenoSphere and the field. I suggest that daily temperatures are compared between environments via RMSE, which calculates the error when considering that PhenoSphere temperatures represent those in the field. I also suggest more integrated indicators, (i) Thermal time, to test if errors accumulate with time, potentially causing a bias in the prediction of developmental stages or, alternatively, if they compensate over time (ii) the number of days with afternoon temperature higher than a threshold, typically 30°C for maize, and with night temperatures lower than a threshold, typically 14°C for maize, to test if extreme events occur with similar frequencies.

I would suggest that the same methods are used to compare light and VPD between PhenoSphere and field.

Response:

Thank you very much for these valuable suggestions. We calculated and included RMSE for the temperature, VPD, and thermal time days comparisons. We treated the field 2016 as a template and all other environments as predicted. We further calculated the daily thermal time contribution of each calendar day (which integrates your request for the upper and lower temperature threshold). The comparison between thermal time days now clearly showed the high similarity (highest correlation, lowest RMSE) between the field 2016 and the 'PhenoSphere 2016 sim' while the correlations for all other combinations are very weak. See l. 132 – 146 and additional Supplemental Figures 2, 3, 4, 5.

4. I was disappointed by the phenotypic variables taken into account for comparing plants in PhenoSphere and in the field. Time to anthesis and plant height are rather crude variables in a phenomic study, for such a sophisticated facility. Furthermore, flowering time is usually well predicted via thermal time in all environments (e.g. multiple experiments in growth chamber, field and greenhouse). It is therefore expected that they match here between PhenoSphere and the field, this is not original. It would be interesting to know to what extent the relation between thermal time and time to anthesis is common to all environments, I expect it will be the case. Similarly, the difference in

the date of maximum growth, which differs between experiments, may be accounted for by differences in thermal time. It would be worth testing this rather than stating that these dates either differ or are similar between experiments.

Response:

Thank you for this hint. We fitted a 4-parameter nonlinear mixed effect model for plant height dependent on the thermal time. The 'PhenoSphere 2016 sim' proved to be a robust simulation of the field 2016 environment even when scaling for temperature. However, we found that the simulation of the average weather 'PhenoSphere avg' lead to the fastest growth (even faster than in the glasshouse) when accounting for temperature by thermal time. See l. 203 – 208 and additional Supplemental Figures 11 & 15.

5. Experiments were carried out with 11 maize lines, but only mean values for all genotypes are presented. It would be important to test to what extent the ranking of genotypes differs or not between environments.

Response:

Thank you for pointing this out. Previously, we did not mention the information on the individual genotypes in the results part. We added a sentence (l. 214 – 217) and pointed the reader to the Supplementary Figure 12 - 15, in which we show the modelled growth trajectory genotype-wise for all traits (now also included for the thermal time dependent model) in every environment.

6. The discussion repeats that PhenoSphere is the missing link between field and controlled conditions, without further arguments than in the introduction. I would expect a more thorough discussion: what can be done in PhenoSphere that cannot be done in a field equipped with sensors? or an equipped greenhouse? Pros and cons should be discussed, e.g. in line with the table proposed by Poorter et al in their paper in New Phytologist 2016 212,838

Response:

Many thanks for this helpful suggestion. We added a paragraph to the discussion (see. l. 304 - 317), in which we referred to Poorter et al., (2016) and explained the limitations of typical growth chambers and glasshouses which are overcome by the technical capabilities of the PhenoSphere, both, above and below ground. Here also restrictions such as the lack of UV-B radiation, the inability to apply frost, and the limited extent of the currently considered interactions of plants with biotic environmental factors are mentioned.

Minor points

Line 97 and below. The term 'modelled' is not appropriate, there is no model here. 'Programmed' would probably be better.

Response:

We agree and changed the term accordingly.

Line 98 and below. Similarly, the word 'variables' should be preferred to 'parameters' for environmental variables. Parameter has a strict meaning in modelling, namely a characteristic that is

specific to an object and does not change with time, e.g. albedo or parameters of a soil water release curve.

Response:

We fully agree and changed the wording accordingly. We also carefully changed “parameter” throughout the rest of the manuscript where appropriate.

Line 161 Differences in date between environment may be accounted for by differences in temperature (they probably are). This can be easily tested via the use of thermal time in all environments, for answering the question ‘is this due to bias in temperature, or to a more physiological reason’.

Response:

Thank you for the suggestion. As mentioned above, we fitted a nonlinear model for plant height dependent on thermal time days and still found no difference for the temporal growth projection of the population in the ‘PhenoSphere 2016 sim’ and the 4 field experiments. When the data were transformed to thermal time days the ‘PhenoSphere avg’ became the environment with the fastest growth, probably due to factors such as light intensity or soil conditions. See l. 203 – 208.

Line 183. I do not understand which variance is meant here

Response:

Thank you for pointing that out. We meant the conditional and marginal R^2 of the linear mixed effect model. We changed the text accordingly (l. 210 – 213).

Line 201. These values of heritability are not very different from those in a well-managed field experiment?

Response:

We would like to point out that these values are not heritability in the sense of doi.org/10.1534/genetics.119.302134, where random effects of replicates or years or locations get weighted for their number, but they are raw variance components of the model. Such as that the agreement repeatability of the fixed and random effects and the residual add up to 1. We agree that in a well-managed field with a footprint of 116 m² the environment should be homogeneous enough that with a well-specified model most variance components could be accounted for. This is why we performed the homogeneity experiment: we wanted to test whether we have a homogeneous environment and do not suffer from some unforeseen bias in the PhenoSphere compartment.

Reviewers' Comments:

Reviewer #1:

Remarks to the Author:

The authors have replied many of the questions addressed by the reviewers, which I really appreciate. However, the key indicator on my view to prove that PhenoSphere mimics field conditions is to provide grain yield estimations. My impression is that authors have tried to skip my question just through the justification that getting yield data was either not scheduled in this study or it will be published in the future.

"We fully agree with the reviewer that comparisons of (grain) yields achieved in field trials and in the PhenoSphere are highly important. After the achievements made in the experiments presented in this manuscript, we therefore initiated further investigations specifically designed for this purpose and look forward to publish these in due course. The goals of the present study were to test, optimize, and validate the ability to elicit growth habits and developmental programs of (crop) plants very similar those under field conditions by exposing them to well-designed growth conditions in the PhenoSphere."

I realize the term "yield" has been mostly removed from the revised version of the manuscript but this is not the issue. Actually, I don't ask even the genotypic range of the yields achieved but just the mean values and then inserting in the Discussion few sentences comparing this yield with the actual yield achieved under field conditions. I may understand the planting density (plants per unit area) is probably far lower than in the field due to logistic reasons (e.g. to allow moving inside the chamber) but even so yield per individual plant or even per row length may give a good indication on how close are the conditions inside this platform to the field. In fact, grain yield of individual plants should be actually higher than those in the field, just because the lower planting density.

Otherwise still I don't see a clear advantage of PhenoSphere from the already existing platforms which, while being probably much less costly for construction and maintenance, have been already delivering very useful information in terms of growth and phenology which may help not only to refine crop models, but also to be directly used as a tool for breeding (e.g. Cabrera-Bosquet et al. 2016 *New Phytol* ; Lacube et al 2017 *PCE*; Perez et al. 2019 *PCE* ; Millet et al. 2019 *Nature Genetics* ; Brichet et al 2017 *Plant Methods* ; Zhanf et al. 2017 *Plant Physiol.*)

Reviewer #2:

Remarks to the Author:

Authors have seriously taken my remarks into account. In particular, I am reassured by the values of light and VPD, which are an essential component of environmental conditions in addition to temperature.

The homogeneity experiment is welcomed

I would suggest to present the data about light and VPD in the main text. As stated earlier, controlling temperature is relatively easy in greenhouses and growth chamber. For me, one of the major achievements of PS is that plants are grown in high light and VPD, not at all common

The discussion is now interesting, because it really consider pro and cons, some ideas are welcomed. However, it is slightly repetitive and too long, so important messages are lost. Authors may consider to shorten it, in particular deleting the first paragraph that brings no new information

'Major' points (not so major)

1- While data on light and VPD are now provided, it is not clear how they are imposed, what is their time course during one day

2- Instead of the somewhat contrived protocol with median days, why not just reproducing a year of interest, chosen in view of a working hypothesis. For instance, no experiment was planned in 2022 so we have no data corresponding to this interesting year. I would dream at reproducing a 2022 experiment and comparing it to 'real' field experiments in other years.

Minor points

Lines 36 and 41. What do you mean by 'relevant' ? An environmental exists per se, it cannot be irrelevant ?

Line 42. I assume that the comparison is with the field, but this is not said

Line 82. 'rate' instead of 'speed'

Lines 102 and 103 : 'modelled' is not the correct word (there is no model here) 'reproduced' or 'simulated' ?

Line 113. I cannot understand why this complicated design was used in the case of a single year, in which you do not need to simulate several situations in a single experiment. Why not just reproducing the climate ?

Line 158 unclear, rephrase (I understand that you wish to see to what extent phenotypes can be reproduced, not study the effect of environment ?

Line 236. The discussion is quite long, consider deleting this first, non informative, paragraph

Line 247. You have shown that environmental conditions in PS acceptably match those in the field, they are still not the "same" as stated here

Line 258. See my second 'major' point

Editor Note: The following text was added after further consultation with the reviewer.

After some thought I would tend to agree with reviewer 1, because the MS hesitates between two objectives

a. HTP phenotyping per se. Many authors, including myself, consider that indoor phenotyping platforms are not for measuring grain yield because environmental conditions and management practices are too different from those in the field. Instead, they serve to measure traits that can hardly be measured in the field, in a range of well characterized conditions (e.g. plant architecture, transpiration rate corrected for evaporative demand, plant-to-plant interactions etc). I stated in my first review that I was disappointed by the traits reported in the MS, which can be measured in the field via imaging with cameras carried by UAV or ground 'phenomobiles', whereas no novel trait was reported. This point was not addressed in the second version of the MS.

b. Reproducing field conditions, which is the objective claimed in the MS. If one fully follows this objective, then the grain yield of studied genotypes should be measured (or at least approached as suggested by reviewer 1), and compared to those in the corresponding field experiment. Furthermore, this objective would merit a better discussion: If one wants to analyse plant behaviour in field conditions, the most straightforward approach is to perform experiments in the field. As stated in my reviews, the main comparative advantage of PhenoSphere is to reproduce very specific conditions that occur one year out of 10 or 20, such as 2022, in which one wants to analyse genotypes without waiting for the right year.

As it is, the MS hesitates between these two objectives, and provides neither (a) novel traits that cannot be measured in the field and may explain yield variations in field experiments, nor (b) a full demonstration of the reproduction of field experiments which would logically involve measurement of grain yield. A fully original article would need to follow either of these two objectives.

Point-by-point response to the reviewers' comments
(line numbers refer to the revised manuscript document with tracked changes)

REVIEWER COMMENTS and RESPONSES

Reviewer #1 (Remarks to the Author):

The authors have replied many of the questions addressed by the reviewers, which I really appreciate. However, the key indicator on my view to prove that PhenoSphere mimics field conditions is to provide grain yield estimations. My impression is that authors have tried to skip my question just through the justification that getting yield data was either not scheduled in this study or it will be published in the future.

Thank you for your remark. We can assure the reviewer that we did not intend to publish yield data of these experiments in a separate paper. Rather, we deemed the seed data to be of low information content (and even confusing), because we meticulously set up the experiments to track in all investigated environments the vegetative phenotypic progression of plants (up to the flowering stage) of a highly diverse population of maize lines. This experimental design is not suitable for a thorough assessment of (seed) yield formation. Nevertheless, we recorded the basic yield parameters of every experiment for the sake of a comprehensive documentation.

“We fully agree with the reviewer that comparisons of (grain) yields achieved in field trials and in the PhenoSphere are highly important. After the achievements made in the experiments presented in this manuscript, we therefore initiated further investigations specifically designed for this purpose and look forward to publish these in due course. The goals of the present study were to test, optimize, and validate the ability to elicit growth habits and developmental programs of (crop) plants very similar those under field conditions by exposing them to well-designed growth conditions in the PhenoSphere.”

I realize the term “yield” has been mostly removed from the revised version of the manuscript but this is not the issue. Actually, I don't ask even the genotypic range of the yields achieved but just the mean values and then inserting in the Discussion few sentences comparing this yield with the actual yield achieved under field conditions. I may understand the planting density (plants per unit area) is probably far lower than in the field due to logistic reasons (e.g. to allow moving inside the chamber) but even so yield per individual plant or even per row length may give a good indication on how close are the conditions inside this platform to the field. In fact, grain yield of individual plants should be actually higher than those in the field, just because the lower planting density.

We realized from the remarks of both reviewers that despite the experimental setup designed for different purpose it is desirable present seed yield data and to discuss the in the manuscript. Thus, we calculated BLUPs of all yield parameters and presented them in a short paragraph in the results part and added a corresponding discussion section. Furthermore we explained the yield data acquisition in the M&M part and added the developed code with the Code Availability, and also

provided the raw measurement data regarding the yield traits.

Otherwise still I don't see a clear advantage of PhenoSphere from the already existing platforms which, while being probably much less costly for construction and maintenance, have been already delivering very useful information in terms of growth and phenology which may help not only to refine crop models, but also to be directly used as a tool for breeding (e.g. Cabrera-Bosquet et al. 2016 New Phytol ; Lacube et al 2017 PCE; Perez et al. 2019 PCE ; Millet et al. 2019 Nature Genetics ; Bricchet et al 2017 Plant Methods ; Zhanf et al. 2017 Plant Physiol.)

We agree with the reviewer that it is interesting to the reader and we need to show that maize plants can fully mature in the PhenoSphere. But we also discuss that for a proper assessment of the (seed) yield of certain genotypes formed in the PhenoSphere upon exposure to a particular environmental regime, conductance of different, specifically designed experiments will be necessary.

Reviewer #2 (Remarks to the Author):

Authors have seriously taken my remarks into account. In particular, I am reassured by the values of light and VPD, which are an essential component of environmental conditions in addition to temperature.

The homogeneity experiment is welcomed

I would suggest to present the data about light and VPD in the main text. As stated earlier, controlling temperature is relatively easy in greenhouses and growth chamber. For me, one of the major achievements of PS is that plants are grown in high light and VPD, not at all common

Thank you for your suggestion. We have now mentioned VPD earlier in the main text (line 71). Furthermore, we have added an explanation about the imposition of the light regimes and relative humidity values in the results part (Line 122 - 128).

The discussion is now interesting, because it really consider pro and cons, some ideas are welcomed. However, it is slightly repetitive and too long, so important messages are lost. Authors may consider to shorten it, in particular deleting the first paragraph that brings no new information

We followed your suggestion and removed the first paragraph of the discussion.

'Major' points (not so major)

1- While data on light and VPD are now provided, it is not clear how they are imposed, what is their time course during one day

We agree and found the explanation in the result part to lack detail. We added information about the imposition of the light regimes and the relative humidity values now also in the results part (Line 122 -128). We regret that we cannot provide the same hourly resolved plots for the light measurement as we did for °C and VPD, because different kind of sensors and data logging systems were used in the field and PhenoSphere environments (180° pyranometer in the field, a PAR sensor with a sensitivity corresponding to the absorption spectrum of chlorophyll in the PhenoSphere, and a LI-COR light meter for light intensity measurements in the PhenoSphere), which prevented a direct comparison. If you insist on an hourly plot we could think about plotting light intensity in each environment relatively. But, to be transparent, we would like to point to the sensor data, which is made available with the manuscript, in which we provide all measured light intensities from the different sensors.

2- Instead of the somewhat contrived protocol with median days, why not just reproducing a year of interest, chosen in view of a working hypothesis. For instance, no experiment was planned in 2022 so we have no data corresponding to this interesting year. I would dream at reproducing a 2022 experiment and comparing it to 'real' field experiments in other years.

We share the reviewer's enthusiasm and would be thrilled to have a system in which we could replicate a single season on a sub-hour scale. However, in the PhenoSphere we went for an approximation of days due to technical limitations as the input slots have a limit and need to be adjusted weekly. Furthermore, the input relies on manual programming. We apologize that we did not explore other single season environments. An experiment in the PhenoSphere is quite expensive and budget constraints did not allow us to explore more iterations. The facility is now continuously used for other research projects.

Minor points

Lines 36 and 41. What do you mean by 'relevant' ? An environmental exists per se, it cannot be irrelevant ?

Thank you, we agree and removed 'relevant' in both instances.

Line 42. I assume that the comparison is with the field, but this is not said

We changed the sentence and properly noted the intended comparison with the field (Line 42)

Line 82. 'rate' instead of 'speed'

Now in line 84, changed accordingly.

Lines 102 and 103 : 'modelled' is not the correct word (there is no model here) 'reproduced' or 'simulated' ?

Now in line 105, we changed it to 'reproduced'.

Line 113. I cannot understand why this complicated design was used in the case of a single year, in which you do not need to simulate several situations in a single experiment. Why not just reproducing the climate ?

Due to a limit of the number of input slots available for programming the environment control system of the PhenoSphere, we decided to approach the single season climate also via a weekly approximation scheme. Theoretically an hourly input would be possible but, we deemed it not to be feasible for an experiment which covers a whole growing season of maize.

Line 158 unclear, rephrase (I understand that you wish to see to what extent phenotypes can be reproduced, not study the effect of environment ?

Thank you, we rephrased this sentence (now in Line 168).

Line 236. The discussion is quite long, consider deleting this first, non informative, paragraph

We agree and removed the paragraph.

Line 247. You have shown that environmental conditions in PS acceptably match those in the field, they are still not the "same" as stated here

Thank you, we rephrased the sentence to be more precise.

Line 258. See my second 'major' point

Please see our comment to your second major point.

After some thought I would tend to agree with reviewer 1, because the MS hesitates between two objectives

a. HTP phenotyping per se. Many authors, including myself, consider that indoor phenotyping platforms are not for measuring grain yield because environmental conditions and management practices are too different from those in the field. Instead, they serve to measure traits that can hardly be measured in the field, in a range of well characterized conditions (e.g. plant architecture, transpiration rate corrected for evaporative demand, plant-to-plant interactions etc). I stated in my first review that I was disappointed by the traits reported in the MS, which can be measured in the field via imaging with cameras carried by UAV or ground 'phenomobiles', whereas no novel trait was reported. This point was not addressed in the second version of the MS.

b. Reproducing field conditions, which is the objective claimed in the MS. If one fully follows this objective, then the grain yield of studied genotypes should be measured (or at least approached as suggested by reviewer 1), and compared to those in the corresponding field experiment. Furthermore, this objective would merit a better discussion: If one wants to analyse plant behaviour in field conditions, the most straightforward approach is to perform experiments in the field. As stated in my reviews, the main comparative advantage of PhenoSphere is to reproduce very specific conditions that occur one year out of 10 or 20, such as 2022, in which one wants to analyse genotypes without waiting for the right year.

As it is, the MS hesitates between these two objectives, and provides neither (a) novel traits that cannot be measured in the field and may explain yield variations in field experiments, nor (b) a full demonstration of the reproduction of field experiments which would logically involve measurement of grain yield. A fully original article would need to follow either of these two objectives.

We followed the remarks of both reviewers and decided to include the properly analyzed seed data which were recorded alongside each experiment in each environment. We presented the yield parameters now in an additional short result paragraph and added a section in the discussion. Furthermore, we updated the M&M part to describe the yield measurement and the required statistics used in the analysis. The developed code is provided and the raw yield data was integrated in the supplemental measurement data sheet.

While the experiments were not designed to determine seed yields of the investigated genotypes, the data show that the PhenoSphere in general is suitable to let maize plants fully mature. But for a proper assessment of the (seed) yield of certain genotypes formed in the PhenoSphere upon exposure to a particular environmental regime, different, specifically designed experiments need to be conducted.

Reviewers' Comments:

Reviewer #2:

Remarks to the Author:

Authors have done their best to address my comments. The added results about yield components are interesting per se, but they should be presented more positively, although critically.

Minor points

1- l 229 : This should be 'yield components' not parameters (these are measurements, not parameters)

2- l 230-232 these lines should be deleted, these are defensive arguments. If you state that PS represents the field, it is natural to present yield components, and then have a critical assessment of these results

3- The following lines are not terribly clear,

- Prefer biological terms (grain number, individual grain weight) rather than agronomic jargon (thousand kernel weight, kernel count)

- Just state the results, potentially with an interpretation why grain number was lower than in the field (probably because of lower intercepted light, see Millet et al 2019)

- At the end of the para, you may want to evaluate the ability of PS to analyze yield based on these results (instead of stating that at the beginning of the para

Point-by-point response to the reviewers' comments
(line numbers refer to the revised manuscript document with tracked changes)

REVIEWERS' COMMENTS and RESPONSE

Reviewer #2 (Remarks to the Author):

Authors have done their best to address my comments. The added results about yield components are interesting per se, but they should be presented more positively, although critically.

Minor points

1- I 229 : This should be 'yield components' not parameters (these are measurements, not parameters)

We followed your suggestion and changed it 'Yield parameters to 'Yield components' and also changed it in all other instances in the manuscript.

2- I 230-232 these lines should be deleted, these are defensive arguments. If you state that PS represents the field, it is natural to present yield components, and then have a critical assessment of these results

We deleted the first sentence as suggested. We added further critical assessment of these results in the discussion.

3- The following lines are not terribly clear,

- Prefer biological terms (grain number, individual grain weight) rather than agronomic jargon (thousand kernel weight, kernel count)
- Just state the results, potentially with an interpretation why grain number was lower than in the field (probably because of lower intercepted light, see Millet et al 2019)
- At the end of the para, you may want to evaluate the ability of PS to analyze yield based on these results (instead of stating that at the beginning of the para

Thank you. We updated all text and figures to use the suggested biological terms.

We added further evaluation of the yield results in the discussion section.

We wrote in line 314:

"Besides the consequences of the wide range of flowering times in the investigated maize population, environmental parameters such as the total intercepted light, or the night temperatures or soil water potential during flowering have been shown to affect grain number in a genotype-dependent manner²² and may have contributed to the differences between the 'PhenoSphere avg' and 'PhenoSphere 2016 sim' experiments. Interestingly, the observed effects on grain yield and yield

components were genotype dependent with different genotype x environment interactions (Supplementary Fig. 22).”

Here we also cited Millet et al 2019 in citation number 22.

And further in line 326: “Furthermore, the aforementioned environmental parameters need to be considered and carefully set in experiments designed to evaluate grain yield in the PhenoSphere.”